# Learning Rate Matters: Vanilla LoRA May Suffice for LLM Fine-tuning

## Abstract

Low-Rank Adaptation (LoRA) is the prevailing approach for efficient large language model (LLM) fine-tuning. Building on this paradigm, recent studies have proposed alternative initialization strategies and architectural modifications, reporting substantial improvements over vanilla LoRA. However, these gains are often demonstrated under fixed or narrowly tuned hyperparameter settings, despite the known sensitivity of neural networks to training configurations. In this work, we systematically re-evaluate four representative LoRA variants alongside vanilla LoRA through extensive hyperparameter searches. Across mathematical and code generation tasks on diverse model scales, we find that different LoRA methods favor distinct learning rate ranges. Crucially, once learning rates are properly tuned, all methods achieve similar peak performance (within 1–2%), with only subtle rank-dependent behaviors. These results suggest that vanilla LoRA remains a competitive baseline and that improvements reported under single training configuration may not reflect consistent methodological advantages. Finally, a second-order analysis attributes the differing optimal learning rate ranges to variations in the largest Hessian eigenvalue, aligning with classical learning theories.

## 1. Introduction

Despite the rapidly growing capabilities of pretrained large language models (LLMs), fine-tuning remains a fundamental step for adapting these models to specialized applications in diverse domains such as medicine (Anisuzzaman et al., 2025) and finance (Djagba & Saley, 2025). However, modern LLMs typically contain billions of parameters, making full-parameter fine-tuning (Full FT) prohibitively

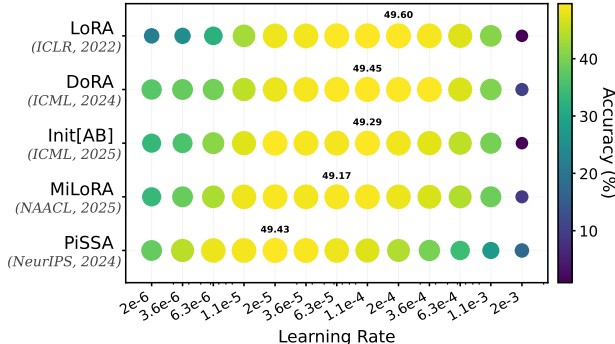

*Figure 1.* Performance of Qwen3-0.6B fine-tuned on mathematical reasoning tasks across learning rates. Different methods reach a similar performance level once the learning rate is properly tuned.

expensive in terms of memory and computation. These constraints have motivated sustained research interest in developing parameter-efficient fine-tuning (PEFT) methods, which allow task-specific learning while updating only a small fraction of parameters.

Even though PEFT methods span diverse design paradigms, ranging from prompt-based approaches (Li & Liang, 2021; Lester et al., 2021) to adapter-based methods (Houlsby et al., 2019; He et al., 2021), low-rank adaptation (LoRA), introduced by Hu et al. (2022), has emerged as the de facto standard. Inspired by the low intrinsic dimensionality observed in pretrained models (Li et al., 2018), Hu et al. (2022) hypothesize that task-specific parameter updates can be well approximated by low-dimensional structures. Consequently, they inject pairs of trainable decomposition matrices into selected layers while keeping the pretrained weights frozen. After training, these learned low-rank adapters can be merged into the original backbone, thereby incurring no additional inference latency.

Even with its popularity, LoRA has been shown to underperform Full FT on challenging tasks in programming and mathematics (Biderman et al., 2024). This gap has in turn spurred recent efforts toward advanced LoRA variants (Zhu & Nguyen, 2025), with promising performance improvements reported. On LLaMA (Touvron et al., 2023), for example, Meng et al. (2024a) presented around a 10% accuracy improvement on GSM8K (Cobbe et al., 2021) by modifying LoRA initialization strategies, while Liu et al.

[1]Anonymous Institution, Anonymous City, Anonymous Region, Anonymous Country. Correspondence to: Anonymous Author <anon.email@domain.com>.

(2024a) reported substantial gains of 37.2% on common-sense reasoning tasks by separately learning magnitude and directional updates of pretrained weight matrices.

Yet, the results in a majority of work along this line were obtained with hyperparameters directly inherited from prior studies, or only fine-tuned in a narrow range. To be specific, in Figure 2, we collect 42 LoRA publications from major AI conferences and journals over the past three years, and include 10 high-impact or recently released preprints, to investigate whether their training involved tuning key hyperparameters—namely, learning rate, batch size, and rank. The statistics clearly reveal that hyperparameter search is not a standard practice in the field, with only one paper simultaneously considering three hyperparameters and fewer than 30% tuning the learning rate, raising questions about the extent to which the reported gains can be attributed to genuine methodological improvements, particularly given the well-known sensitivity of neural networks to training configurations (LeCun et al., 2002; Bengio, 2012). This is especially critical when it comes to LoRA on LLMs, where careful learning rate tuning is demonstrated to be essential for eliciting strong performance, and optimal settings are contingent on both the base model and the target problem (Biderman et al., 2024; Schulman & Lab, 2025; Yan et al., 2025).

To address above concern, we select four representative advanced LoRA variants and conduct a large-scale hyperparameter search, benchmarking them against vanilla LoRA in a head-to-head manner. Under a unified evaluation protocol, we surprisingly find that once the learning rate is properly tuned, all methods peak at similar performance levels, exhibiting no systematic advantages over the vanilla LoRA. For example, in Figure 1, where learning rates are tuned across three orders of magnitude for Qwen (Yang et al., 2025) with rank 128, all methods achieve accuracies within a narrow 0.43% range. Moreover, different methods operate under disparate learning rate ranges (e.g., a 10× difference between PiSSA (Meng et al., 2024a) and LoRA in Figure 1), suggesting that success under a single training configuration cannot be taken as evidence of robust and reliable improvements. This phenomenon is not isolated; we consistently observe such performance parity across diverse models, tasks, and LoRA ranks. Notably, within these marginal performance variations, rank-dependent behaviors emerge: some advanced variants may slightly outperform LoRA at higher ranks while lagging behind at lower ones (or vice versa), highlighting the importance of verifying improvements across the entire rank spectrum.

By delving into the fundamentals in learning theories (LeCun et al., 1992; Lewkowycz et al., 2020), we provide an explanation for the importance of fine-tuning learning rates during LoRA finetuning and uncover the reasons be-

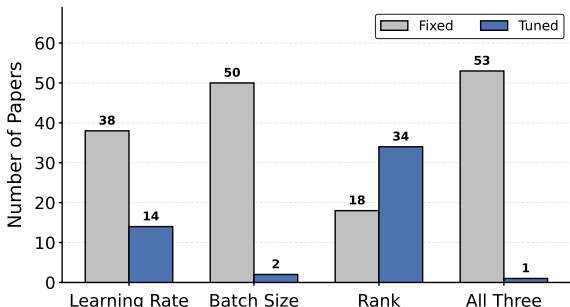

*Figure 2.* Frequency of advanced LoRA-based PEFT studies, categorized by whether learning rate or batch size tuning was applied and whether comparisons with vanilla LoRA across different ranks were conducted. Crucially, only one of the surveyed works simultaneously covered all three dimensions. Refer to Appendix Sec. A for detailed data counts.

hind different desirable learning rate ranges among various LoRA finetuning methods. Specifically, we demonstrate that PiSSA exhibits a significantly larger maximum Hessian eigenvalue compared to vanilla LoRA, which theoretically justifies its requirement for a lower learning rate. In summary, the main contributions of this work are:

- We demonstrate that while different methods require distinct optimal learning rate ranges, they yield comparable performance when configured to their optimal settings. For example, on Qwen3-0.6B, the top-performing method (LoRA) leads the runner-up (DoRA) by only 0.15%, and the least effective method (MiLoRA) by 0.43% (cf. Figure 1).

- We identify intriguing rank-dependent behaviors. For example, MiLoRA demonstrates some improvement over vanilla LoRA when the rank is 8 on the mathematical reasoning task with Gemma, yet slightly underperforms LoRA when the rank is 256.

- By analyzing the eigenvalues of the Hessian across LoRA variants, we find that the optimal learning rate is generally negatively correlated with the maximum eigenvalue, aligning with classical learning theories.

## 2. Related Work

### 2.1. Systematic Empirical Re-evaluation of Prior Claims

Incomplete performance evaluation remains a persistent concern (Sculley et al., 2018; Lipton & Steinhardt, 2019). For instance, Melis et al. (2017) revealed that two published improvements to the vanilla LSTM (Graves, 2012), originally attributed to complex network designs, were in fact due to more careful hyperparameter tuning. Under a fair tuning protocol, the standard LSTM emerged as the best-performing

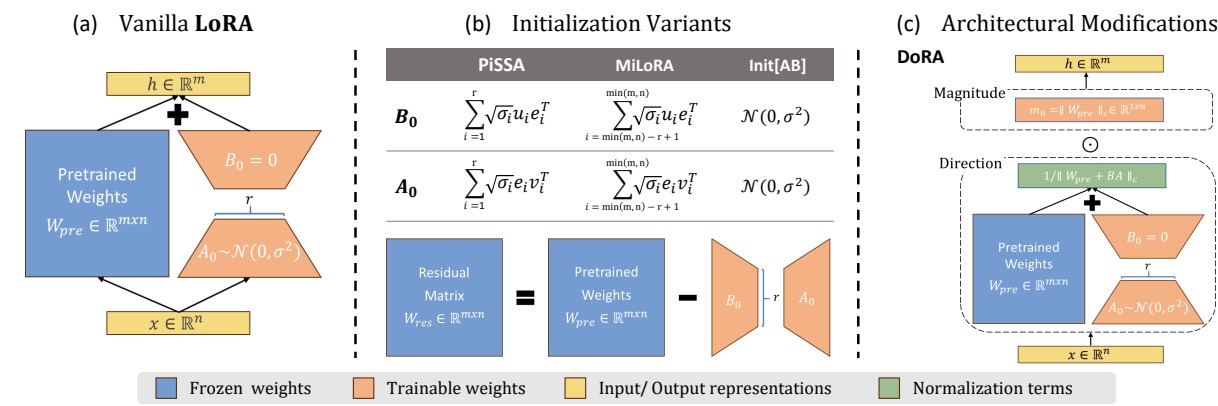

*Figure 3.* Overview of our considered PEFT methods: (a) Vanilla LoRA (Sec. 3.1), (b) three representative initialization variants (Sec. 3.2), and (c) one architectural modification (Sec. 3.3).

architecture. In the same vein, Lin et al. (2023) pointed out that simple baselines such as linear SVM (Boser et al., 1992) are competitive with BERT (Devlin et al., 2019)-based methods for text classification, sometimes even outperforming them with a clear gap.

Such systematic empirical re-evaluation has also been done across diverse machine learning subfields. Examples include not only traditional topics such as image classification (Chatfield et al., 2014), graph neural networks (Shchur et al., 2018), generative adversarial networks (Lucic et al., 2018), recommender systems (Ferrari Dacrema et al., 2019), metric learning (Musgrave et al., 2020), and neural network pruning (Blalock et al., 2020), but also more recent areas like optimizers (Schmidt et al., 2021), reinforcement learning (Eimer et al., 2023), preference optimization (Ahrabian et al., 2025), and model merging (Hitit et al., 2025).

While empirical studies benchmarking LoRA with other PEFT methods such as prefix tuning (Li & Liang, 2021) and BitFit (Zaken et al., 2022) exist (He et al., 2021; Hu et al., 2023; Zheng et al., 2024b; Männistö et al., 2025; Belanec et al., 2025), few studies specifically focus on comparing LoRA and its advanced variants. More concerning, training hyperparameters in many of these works were kept fixed without method-specific optimization. Therefore, practitioners are left without clear and reliable guidance when choosing LoRA-based methods.

### 2.2. LoRA Hyperparameter Tuning

Theories regarding LoRA's lack of Lipschitz smoothness (Sun et al., 2024; Malinovsky et al., 2024) and its spurious loss landscape (Liu et al., 2025) point toward its intrinsic sensitivity to hyperparameter variations. Consequently, many research efforts have been invested in finding optimal training setups, such as learning rate (Hayou et al., 2024b), rank (Zhang et al., 2025a), initializations (Hayou et al., 2024a), scaling factor (Kalajdzievski, 2023), dropout (Lin

et al., 2024), and adapter placements (Fomenko et al., 2024; Hayou et al., 2025). Despite these insights into individual hyperparameters, establishing unified configuration guidelines remains an ongoing pursuit. For example, recent works have sought to derive practical "rules of thumb" through extensive, joint evaluations across multiple hyperparameter dimensions (Biderman et al., 2024; Schulman & Lab, 2025). Addressing the computational bottleneck of such extensive searches, Yan et al. (2025) took a system-level approach, optimizing hardware resources to maximize training throughput specifically for LoRA hyperparameter tuning.

Despite these efforts to optimize LoRA hyperparameters, few studies have examined whether LoRA and its variants require distinct hyperparameter settings. Most relevant to our work is Zhang et al. (2025h), which provided a helpful theoretical framework revealing the interplay between LoRA initialization strategies and hyperparameters such as learning rate and scaling factor. Specifically, Zhang et al. (2025h) noted an intriguing phenomenon where LoRA and two of its initialization variants, PiSSA (Meng et al., 2024a) and MiLoRA (Wang et al., 2024b), exhibit performance shifts across learning rates. However, this analysis was limited to two learning rates ($2e^{-4}$ and $2e^{-5}$) and the observed trends were further shown to depend on the rank configuration (cf. Zhang et al. (2025h, Appendix G, Table 5-6)). In contrast, our work expands this investigation to a broader set of recent LoRA variants and conducts comprehensive multivariate hyperparameter tuning to identify the best-performing configuration for each method. Moreover, we also provide intuitions of the underlying factors driving the observed performance differences and trends.

## 3. Background: LoRA PEFT Methods

In this section, we introduce background and notations for five selected PEFT methods, including LoRA and its representative improvements, i.e., PiSSA (Meng et al., 2024a),

MiLoRA (Wang et al., 2024b), Init[AB] (Li et al., 2025), and DoRA (Liu et al., 2024a) (Figure 3).

### 3.1. Low-Rank Adaptation

Given a pretrained neural network layer parameterized by $W_{\text{pre}} \in \mathbb{R}^{m \times n}$, LoRA introduced two trainable matrices: the down-projecting $A \in \mathbb{R}^{r \times n}$ and the up-projecting $B \in \mathbb{R}^{m \times r}$ ($r \ll \min(m,n)$). For layer input $x \in \mathbb{R}^n$, the output $h \in \mathbb{R}^m$ is computed as:

$$h = W_{\text{pre}}x + \gamma_r BAx,$$

where $\gamma_r = \frac{\alpha}{r}$ serves as a rank-dependent scaling factor with $\alpha$ being a tunable hyperparameter. At initialization, the two trainable matrices $B$ and $A$ are set to $B_0 = 0$ and $A_0 \sim \mathcal{N}(0, \sigma^2)$ (i.e., Kaiming initialization (He et al., 2015)), respectively, ensuring that fine-tuning starts exactly from the pretrained checkpoint.

### 3.2. Initialization Variants

A line of work has explored better initialization strategies for LoRA. For example, PiSSA and MiLoRA leverage the singular value decomposition (SVD) of $W_{\text{pre}}$ to inform the initialization of $B$ and $A$, while Init[AB] theoretically examines different zero initialization schemes.

**PiSSA.** Aiming to address the potential slow convergence of LoRA, Meng et al. (2024a) proposed to initialize $BA$ with the top-$r$ principal components of the pretrained weight matrix and showed that this approach achieves faster convergence with loss and gradient norm curves similar to those of full fine-tuning. Suppose $W_{\text{pre}}$ admits SVD into $\sum_i \sigma_i u_i v_i^T$ where $\sigma_i$ are singular values in descending order, the LoRA adapter is initialized as:

$$B_0 A_0 = \sum_{i=1}^{r} \sigma_i u_i v_i^T,$$

with

$$B_0 = \sum_{i=1}^{r} \sqrt{\sigma_i} u_i e_i^T, \quad A_0 = \sum_{i=1}^{r} \sqrt{\sigma_i} e_i v_i^T, \quad (1)$$

where $e_i \in \mathbb{R}^r$ denotes the $i$-th standard basis vector. To start fine-tuning from the pretrained weights, the *residual matrix* is defined as

$$W_{\text{res}} = W_{\text{pre}} - B_0 A_0 = \sum_{i=r+1}^{\min(m,n)} \sigma_i u_i v_i^T,$$

and the forward pass becomes:

$$h = W_{\text{res}}x + \gamma_r BAx.$$

**MiLoRA.** Concurrent to PiSSA, Wang et al. (2024b) proposed an analogous approach targeting adaptation to new tasks while maximally retaining the pretrained model's knowledge. Instead of using principal components, MiLoRA initializes the low-rank adapters using bottom-$r$ minor components:

$$B_0 A_0 = \sum_{i=\min(m,n)-r+1}^{\min(m,n)} \sigma_i u_i v_i^T,$$

with $B_0$ and $A_0$ defined analogously to Eq. 1. The residual matrix retains principal components:

$$W_{\text{res}} = W_{\text{pre}} - B_0 A_0 = \sum_{i=1}^{\min(m,n)-r} \sigma_i u_i v_i^T.$$

Wang et al. (2024b) showed experimentally that MiLoRA achieves superior downstream performance with less catastrophic forgetting.

**Init[AB].** Several works have theoretically analyzed the initialization strategies of LoRA (Hayou et al., 2024a; Xu et al., 2025). In particular, Hayou et al. (2024a) confirmed that Init[A] (i.e., randomly initializing $A$ only, the default LoRA setting) generally leads to better performance than Init[B] (randomly initializing $B$ only). Li et al. (2025) further showed that initializing both matrices (i.e., Init[AB]) can provide even better advantage by balancing stability, training efficiency, and hyperparameter robustness. Specifically, Init[AB] initializes both matrices as $B_0 \sim \mathcal{N}(0, \sigma^2)$ and $A_0 \sim \mathcal{N}(0, \sigma^2)$. Since $B_0 A_0 \neq 0$ in this case, the residual matrix $W_{\text{res}}$ is similarly introduced and utilized.

### 3.3. Architectural Modifications

Besides investigating initialization strategies, an even larger body of literature has focused on architecture-level improvements, e.g., VeRA (Kopiczko et al., 2023b), BOFT (Liu et al., 2023b), and GraLoRA (Jung et al., 2025) available in the PEFT library (Mangrulkar et al., 2022).

**DoRA.** Liu et al. (2024a) proposed to learn *magnitude* and *directional* updates of $W_{\text{pre}}$ separately. Formally, the modified forward pass becomes:

$$h = \gamma_r \left( \frac{m}{\|W_{\text{pre}} + BA\|_c} \odot (W_{\text{pre}} + BA) \right) x, \quad (2)$$

where $B$ and $A$ are responsible for directional updates and initialized as vanilla LoRA, while $m \in \mathbb{R}^{1 \times n}$ is an additional trainable magnitude vector initialized with $m_0 = \|W_{\text{pre}}\|_c$. Note that $\|.\|_c$ denotes taking the column-wise norm of a matrix, while $\odot$ denotes element-wise multiplication with broadcasting across columns. With only a

marginal increase in trainable parameters introduced by the vector $m$, DoRA has been shown to consistently outperform LoRA, especially in regimes where the rank is small. We select DoRA among architectural variants as it maintains a parameter efficiency similar to vanilla LoRA, unlike the significant reduction in VeRA (from $(m + n)r$ to $m + r$ per layer), and does not require setting additional architectural hyperparameters (e.g., $(m, b)$ in BOFT or $k$ in GraLoRA).

## 4. Learning Rate Matters, Really

### 4.1. Motivation

For the trainable LoRA parameters across layers, collectively denoted as $\theta$, the update rule of Stochastic Gradient Descent (SGD) at step $t$ is:

$$\boldsymbol{\theta}_{t+1} = \boldsymbol{\theta}_t - \eta \mathbf{g}(\boldsymbol{\theta}_t),$$

where $\eta$ is the learning rate. While setting $\eta$ too large causes the optimization step to overshoot, leading to instability or divergence, a value that is too small is insufficient to escape suboptimal local minima or affect convergence rate. To analyze this formally, consider the local geometry characterized by the gradient $\mathbf{g}(\boldsymbol{\theta}_t) \triangleq \nabla \mathcal{L}(\boldsymbol{\theta}_t)$ and the Hessian $\mathbf{H}(\boldsymbol{\theta}_t) \triangleq \nabla^2 \mathcal{L}(\boldsymbol{\theta}_t)$, where $\mathcal{L}$ denotes the loss function. According to classical learning theories (LeCun et al., 1992), the optimal learning rate $\eta^*$ for efficient learning is intrinsically tied to the curvature of the loss landscape at $\boldsymbol{\theta}$, typically scaling inversely with the Hessian's maximum eigenvalue:

$$\eta^* \propto \frac{1}{\lambda_{\max}(\mathbf{H}(\boldsymbol{\theta}))}. \tag{3}$$

Crucially, LoRA initialization variants establish specific training starting points $\boldsymbol{\theta}_0$, resulting in distinct $\mathbf{g}(\boldsymbol{\theta}_0)$, $\mathbf{H}(\boldsymbol{\theta}_0)$, and subsequent training trajectories compared to vanilla LoRA. Therefore, different methods theoretically require their respective calibrations of $\eta$ to ensure efficient convergence, motivating our decision to perform learning rate tuning for a fair and reliable head-to-head comparison across methods.

### 4.2. Experimental Setup

Since model choices, training configurations, and dataset partitioning vary across papers, we establish a unified experimental framework that accommodates all methods fairly. We describe the key components below, with additional implementation details deferred to Appendix B.

**Pretrained Models.** We consider three decoder-only models spanning different scales: Qwen3-0.6B (Yang et al., 2025), Gemma-3-1B (Team et al., 2025), and Llama-2-7B (Touvron et al., 2023). This selection includes both recently released models (Qwen3, Gemma-3) and an older

one (Llama-2) that is widely used in prior art on this subject, enabling us to validate results across LLMs with diverse pretrained capabilities.

**Fine-tuning Tasks.** We train models on two canonical tasks: mathematical reasoning and code generation. For these tasks, we follow the setup of Meng et al. (2024a). In particular, for mathematical reasoning, we use MetaMathQA (Yu et al., 2023) during the training, and GSM8K (Cobbe et al., 2021) and MATH (Hendrycks et al., 2021) for testing; for code generation, we use CodeFeedback (Zheng et al., 2024a) during the training, and HumanEval (Chen, 2021) and MBPP (Austin et al., 2021) for testing. Unless otherwise specified, we report mean accuracy (equivalently, pass@1 for code generation) on the two testing datasets. We also adopt the same data preprocessing: MetaMathQA is subsampled from 395k to 100k samples, and CodeFeedback is filtered to Python-only questions (from 156k to 104k samples).

**Hyperparameter Settings.** We consider batch sizes ($B$) in {16, 64, 128} and ranks ($r$) in {4, 8, 16, 32, 64, 128, 256}. The learning rates ($\eta$) are swept uniformly on a logarithmic scale from $10^{-3}$ to $10^{-6}$, with four values per order of magnitude: $1.1247 \times 10^*$, $2.0000 \times 10^*$, $3.5566 \times 10^*$, $6.3246 \times 10^*$, yielding up to 16 grid points for the learning rate alone. To maintain the computational feasibility of this study, batch size and rank are tuned only for selected model-task combinations; conversely, learning rates are tuned across all combinations, with ranges defined to ensure inclusion of optimal performance. See Appendix Table 3 for a summary of models, tasks, and their corresponding hyperparameter search ranges. Other configurations, such as epoch, adapter placement, and learning rate scheduler, remain fixed across all experiments and are listed in Appendix Table 4. Specifically for the scaling factor $\gamma_r$, we follow Meng et al. (2024a) by setting $\alpha$ equal to $r$ in all our experiments. This results in $\gamma_r = 1$ for all $r$, effectively factoring out the need to tune this hyperparameter (refer to Appendix Sec. C for further discussion).

### 4.3. Results and Observations

We begin by discussing results under fixed rank $r = 128$ in Sec. 4.3.1, and then in Sec. 4.3.2, we analyze how the methods perform across different ranks.

#### 4.3.1. SIMILAR PERFORMANCE LEVELS

Analogous to the hyperparameter search results shown in Figure 1 for Qwen3-0.6B, Table 1 and Figure 4 present the results for Gemma-3-1B and Llama-2-7B. Through comprehensive hyperparameter searches, we consistently observe across different model scales and tasks that these methods peak at performance levels similar to vanilla LoRA. Specif-

| Methods | Batch Size | Learning Rate | | | | | | | | | | | |
|---|---|---|---|---|---|---|---|---|---|---|---|---|---|
| | | 1.1e-5 | 2e-5 | 3.6e-5 | 6.3e-5 | 1.1e-4 | 2e-4 | 3.6e-4 | 6.3e-4 | 1.1e-3 | 2e-3 | 3.6e-3 | 6.3e-3 |
| LoRA | 16 | $9.78_{\pm0.36}$ | $11.16_{\pm0.28}$ | $13.58_{\pm0.18}$ | $15.48_{\pm0.15}$ | $18.43_{\pm0.14}$ | $\mathbf{20.00}_{\pm0.26}$ | $19.93_{\pm0.65}$ | $17.99_{\pm0.55}$ | $11.71_{\pm0.49}$ | $1.52_{\pm0.19}$ | $1.27_{\pm0.59}$ | $1.07_{\pm0.27}$ |
| | 64 | $6.88_{\pm0.04}$ | $9.12_{\pm0.39}$ | $10.79_{\pm0.37}$ | $13.23_{\pm0.25}$ | $15.65_{\pm0.57}$ | $17.54_{\pm0.29}$ | $19.73_{\pm0.16}$ | $\mathbf{20.46}_{\pm0.79}$ | $19.83_{\pm0.91}$ | $13.33_{\pm0.81}$ | $1.48_{\pm0.48}$ | $0.00_{\pm0.00}$ |
| | 128 | $5.70_{\pm0.34}$ | $6.95_{\pm0.23}$ | $9.41_{\pm0.44}$ | $11.43_{\pm0.40}$ | $13.68_{\pm0.77}$ | $15.92_{\pm0.45}$ | $18.58_{\pm0.44}$ | $19.60_{\pm0.09}$ | $\mathbf{20.32}_{\pm0.28}$ | $16.95_{\pm2.70}$ | $0.09_{\pm0.16}$ | $0.00_{\pm0.00}$ |
| DoRA | 16 | $9.89_{\pm0.24}$ | $11.16_{\pm0.51}$ | $13.84_{\pm0.41}$ | $15.61_{\pm0.11}$ | $18.21_{\pm0.45}$ | $20.11_{\pm0.26}$ | $\mathbf{20.96}_{\pm0.57}$ | $18.34_{\pm0.20}$ | $11.90_{\pm0.29}$ | $4.89_{\pm0.99}$ | $0.93_{\pm0.12}$ | $1.16_{\pm0.15}$ |
| | 64 | $6.72_{\pm0.09}$ | $9.19_{\pm0.19}$ | $10.53_{\pm0.20}$ | $13.45_{\pm0.31}$ | $15.72_{\pm0.32}$ | $17.66_{\pm0.20}$ | $19.96_{\pm0.05}$ | $\mathbf{20.82}_{\pm0.32}$ | $19.87_{\pm0.91}$ | $13.53_{\pm1.64}$ | $1.52_{\pm0.45}$ | $0.34_{\pm0.23}$ |
| | 128 | $5.55_{\pm0.11}$ | $7.21_{\pm0.18}$ | $9.72_{\pm0.17}$ | $11.58_{\pm0.25}$ | $13.98_{\pm0.33}$ | $16.19_{\pm0.46}$ | $18.25_{\pm0.23}$ | $19.67_{\pm0.71}$ | $\mathbf{20.33}_{\pm0.64}$ | $12.86_{\pm10.03}$ | $0.13_{\pm0.23}$ | $0.02_{\pm0.03}$ |
| Init[AB] | 16 | $9.73_{\pm0.35}$ | $12.10_{\pm0.14}$ | $14.41_{\pm0.49}$ | $16.73_{\pm0.37}$ | $18.38_{\pm0.53}$ | $20.39_{\pm0.38}$ | $\mathbf{20.55}_{\pm0.40}$ | $18.34_{\pm0.48}$ | $11.94_{\pm0.31}$ | $1.48_{\pm0.24}$ | $1.16_{\pm0.31}$ | $1.45_{\pm0.17}$ |
| | 64 | $6.51_{\pm0.22}$ | $9.15_{\pm0.12}$ | $11.28_{\pm0.20}$ | $13.20_{\pm0.24}$ | $15.88_{\pm0.39}$ | $17.89_{\pm0.30}$ | $20.08_{\pm0.26}$ | $\mathbf{20.98}_{\pm0.33}$ | $19.31_{\pm0.75}$ | $13.97_{\pm0.03}$ | $2.74_{\pm3.83}$ | $0.07_{\pm0.12}$ |
| | 128 | $6.06_{\pm0.35}$ | $7.05_{\pm0.33}$ | $9.53_{\pm0.22}$ | $11.81_{\pm0.08}$ | $13.98_{\pm0.79}$ | $16.46_{\pm0.39}$ | $18.36_{\pm0.21}$ | $20.37_{\pm0.39}$ | $\mathbf{20.66}_{\pm0.39}$ | $17.85_{\pm0.84}$ | $4.40_{\pm7.46}$ | $0.00_{\pm0.00}$ |
| MiLoRA | 16 | $12.44_{\pm0.07}$ | $13.77_{\pm0.25}$ | $16.28_{\pm0.24}$ | $18.45_{\pm0.47}$ | $20.04_{\pm0.19}$ | $\mathbf{20.63}_{\pm0.67}$ | $19.40_{\pm0.80}$ | $15.72_{\pm0.49}$ | $10.22_{\pm0.42}$ | $2.03_{\pm0.95}$ | $1.35_{\pm0.43}$ | $1.56_{\pm0.65}$ |
| | 64 | $8.82_{\pm0.40}$ | $11.25_{\pm0.20}$ | $13.16_{\pm0.11}$ | $15.54_{\pm0.29}$ | $17.43_{\pm0.24}$ | $19.56_{\pm0.33}$ | $\mathbf{20.03}_{\pm0.59}$ | $19.60_{\pm0.78}$ | $17.93_{\pm0.90}$ | $13.65_{\pm0.07}$ | $4.97_{\pm0.40}$ | $0.00_{\pm0.00}$ |
| | 128 | $7.32_{\pm0.33}$ | $9.57_{\pm0.24}$ | $11.76_{\pm0.33}$ | $13.54_{\pm0.12}$ | $16.02_{\pm0.16}$ | $18.39_{\pm0.26}$ | $19.70_{\pm0.34}$ | $\mathbf{19.99}_{\pm0.66}$ | $19.53_{\pm0.47}$ | $16.83_{\pm0.73}$ | $7.45_{\pm1.00}$ | $0.57_{\pm0.81}$ |
| PiSSA | 16 | $14.30_{\pm0.18}$ | $16.10_{\pm0.27}$ | $18.31_{\pm0.12}$ | $19.90_{\pm0.21}$ | $\mathbf{20.61}_{\pm0.28}$ | $19.09_{\pm0.20}$ | $16.10_{\pm0.64}$ | $13.25_{\pm0.55}$ | $8.41_{\pm0.13}$ | $4.67_{\pm0.29}$ | $2.50_{\pm1.27}$ | $0.96_{\pm0.15}$ |
| | 64 | $11.11_{\pm0.05}$ | $13.67_{\pm0.17}$ | $15.56_{\pm0.33}$ | $18.11_{\pm0.23}$ | $19.52_{\pm0.48}$ | $\mathbf{20.68}_{\pm0.77}$ | $20.59_{\pm0.32}$ | $19.11_{\pm0.86}$ | $15.53_{\pm0.37}$ | $9.57_{\pm0.72}$ | $5.78_{\pm0.37}$ | $0.33_{\pm0.46}$ |
| | 128 | $9.42_{\pm0.38}$ | $11.80_{\pm0.28}$ | $14.40_{\pm0.11}$ | $16.23_{\pm0.38}$ | $18.60_{\pm0.21}$ | $19.61_{\pm0.44}$ | $\mathbf{20.65}_{\pm0.44}$ | $19.21_{\pm1.15}$ | $16.91_{\pm0.19}$ | $13.87_{\pm0.97}$ | $6.28_{\pm0.49}$ | $1.19_{\pm0.36}$ |

*Table 1.* Performance of Gemma-3-1B on mathematical reasoning task across varying batch sizes and learning rates ($r = 128$). Results are reported as mean $\pm$ standard deviations over three independent runs. Best results are highlighted in **bold**, and configurations achieving $\geq 18.5\%$ accuracy (i.e., $\approx 90\%$ of the maximum) are shaded in green ( ). While all methods achieve comparable peak accuracies, the optimal learning rates vary depending on both the fine-tuning method and batch size.

ically, the performance gap across all methods is small: 0.99% for Gemma on math, and 0.43% and 1.75% for Llama on math and code, respectively. It is also important to note that while peak performance is similar, the optimal learning rates vary. In particular, PiSSA requires a lower learning rate compared to vanilla LoRA across all model-task combinations, while other methods fall within a similar range to LoRA, typically within the same order of magnitude.

Beyond the optimal learning rates, a closer inspection of the full learning rate spectrum reveals intriguing method-specific behaviors. For instance, we observe that PiSSA tends to remain effective at larger learning rates where other methods diverge: in Figure 4 at $\eta = 1.1 \times 10^{-3}$, PiSSA maintains accuracies of 27.83% and 26.90% on math and code tasks, respectively, while other methods collapse to near-zero performance on at least one of the tasks.

Note that in Table 1, the joint optimization of learning rate and batch size indicates that tuning the learning rate is significantly more critical than tuning the batch size for obtaining best performance in both LoRA and its variants, consistent with early findings for neural networks (Bengio, 2012). For example, with DoRA, fixing the learning rate at $2 \times 10^{-5}$ and tuning only the batch size yields a suboptimal maximum accuracy of 11.16%. In contrast, by fixing the batch size to any value in $\{16, 64, 128\}$ and tuning the learning rate, the model achieves substantially higher performance within the range of 20.5–21.0%. Moreover, we observe that the optimal learning rate scales proportionally with batch size, aligning with the "scaling rule" established in SGD literature (Goyal et al., 2017; Hoffer et al., 2017). This

offers a plausible explanation for the discrepancy between our results and those of Schulman & Lab (2025), where LoRA was reported to suffer from performance degradation with a larger batch size of 128; we hypothesize that this degradation likely stems from a suboptimal learning rate choice. While recent work has highlighted more nuanced batch size effects (e.g. Marek et al. (2025) demonstrated that smaller batch sizes can be beneficial if the learning rate and optimizer hyperparameters are jointly tuned), we leave the extended study for future work.

We further validate our findings across varying number of training samples in Appendix D.1. More numerical results and evaluations of additional model-task combinations, and example model responses are provided in Appendix E and F, respectively.

### 4.3.2. PERFORMANCE COMPARISON ACROSS RANKS

Next, we extend our analysis by varying adapter ranks for Gemma-3-1B, as shown in Figure 5. The results indicate that the previously observed performance parity persists across a wide range of rank settings, with the maximum performance differences among methods being only 2.15% (Math, $r = 8$) and 1.67% (Code, $r = 32$). Interestingly, however, we observe the relative performance of variants compared to LoRA fluctuates across different ranks within these margins.

In particular, PiSSA initially underperforms vanilla LoRA before gradually overtaking it as the rank increases. Taking the math task as an example (Figure 5a), PiSSA exhibits performance deficits of up to 1.67% at low ranks ($r \leq 32$),

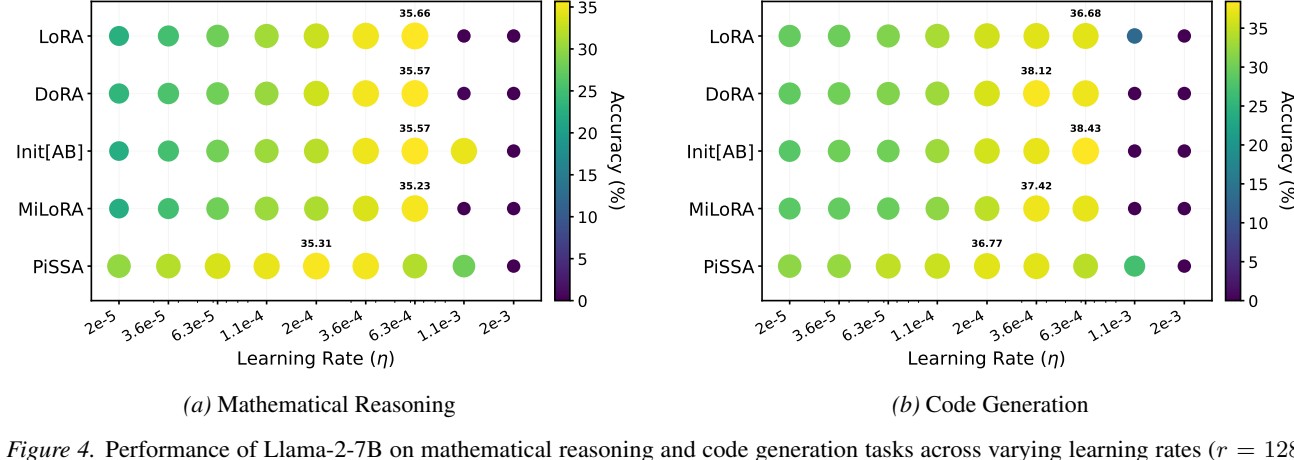

*(a)* Mathematical Reasoning        *(b)* Code Generation

*Figure 4.* Performance of Llama-2-7B on mathematical reasoning and code generation tasks across varying learning rates ($r = 128$, $B = 128$). Notably, PiSSA peaks at lower learning rates but remains effective at larger learning rates on both tasks (e.g., $1.1 \times 10^{-3}$), where other methods diverge.

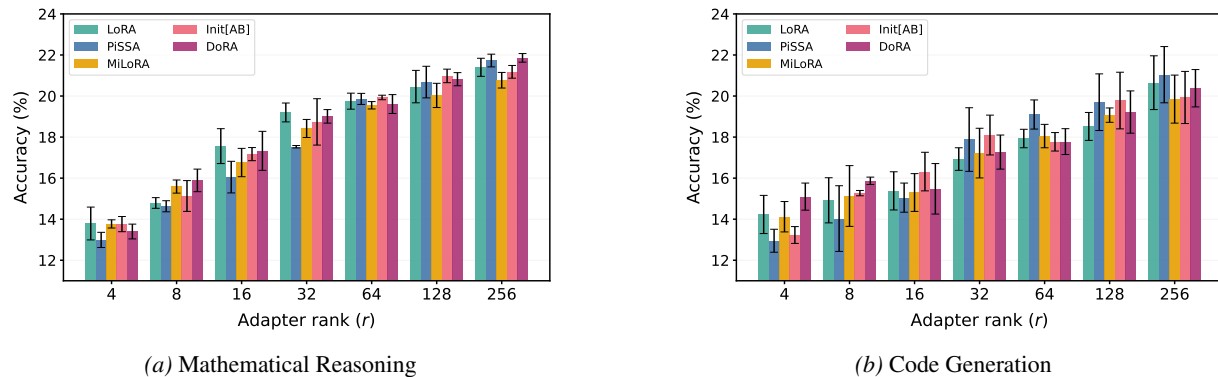

*(a)* Mathematical Reasoning        *(b)* Code Generation

*Figure 5.* Best achievable performance of LoRA and its advanced variants across adapter ranks on Gemma-3-1B ($B = 64$). With properly tuned learning rates, all methods exhibit similar performance improvement trends as the rank increases, though subtle rank-dependent behaviors emerge. Results are reported with means and standard deviations over three independent runs.

but narrows the gap to within 0.11% at $r = 64$ and shifts to a slight gain of 0.65% and 0.33% at $r = 128$ and 256, respectively. In contrast, MiLoRA shows an opposite trend, where it tends to outperform vanilla LoRA at lower ranks but fails to sustain this advantage as the rank increases: it achieves a 0.8% gain at $r = 8$ and aligns with LoRA at $r = 64$, before eventually falling behind by 0.43% and 0.63% at $r = 128$ and 256, respectively. Figure 5b indicates that this rank-dependent dynamics extend to the coding task.

In the case of Init[AB], we observe that it tends to outperform LoRA at medium ranks, e.g., achieving maximal gains of 0.52% on math at $r = 128$ and 1.26% on code at $r = 32$. Yet, the success does not translate to either lower or higher rank scenarios, where Init[AB] typically performs similarly to vanilla LoRA. Turning to DoRA, we observe performance gains against LoRA specifically in low-rank regimes, peaking at 1.1% on math and 0.95% on code at $r = 8$. While this aligns with the claim that DoRA excels especially in low-rank scenarios, we highlight that the improvements are

smaller than those reported in the original work (cf. Liu et al. (2024a, Figure 5)), underscoring the critical importance of learning rate tuning. Furthermore, DoRA's success does not translate uniformly to other low-rank settings (e.g., math at $r = 4$), and it is also worth noting that the parameter overhead introduced by the magnitude vector ($m$ in Eq.2) becomes non-negligible relative to the total trainable parameter counts when the rank is small. Similar performance comparisons across ranks were also conducted on Llama, with results deferred to Appendix Sec. D.2.

## 5. Understanding the Optimal Learning Rate via Hessian Analysis

### 5.1. Sharpness-Learning Rate Relationship

The Hessian of the loss function has been the subject of numerous studies. Geometrically, its top eigenvalue (denoted as $\lambda_{\max}$ for brevity) at a given point represents the maximal curvature of the loss landscape along any direction, com-

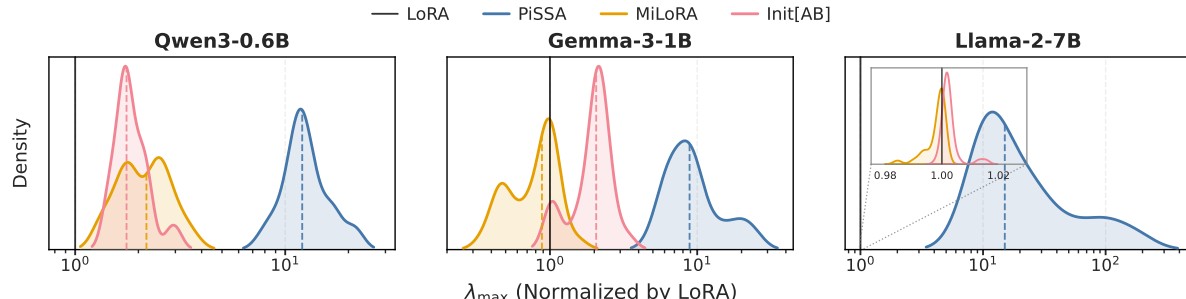

*Figure 6.* Distributions of the ratios of the top loss Hessian eigenvalues relative to LoRA for Query projection matrices across Transformer layers. Dashed lines indicate the medians. With an adapter rank of $r = 128$, PiSSA consistently exhibits the largest eigenvalues. In contrast, MiLoRA and Init[AB] exhibit eigenvalues slightly larger than those of LoRA on Qwen, yet remain comparable to LoRA on Gemma and Llama. Detailed $\lambda_{\max}$ values are provided in Appendix Sec. G.2.

monly referred to as sharpness (Dinh et al., 2017; Lyu et al., 2022; Luo et al., 2024). This metric is closely linked to the optimal learning rate, a connection that originates from the Gauss-Newton method for convex optimization and further elucidated by LeCun et al. (1992) in the context of neural networks. Specifically, it was shown that an efficient learning rate theoretically falls within $1/\lambda_{\max} \leq \eta^* < 2/\lambda_{\max}$ under quadratic approximation, whereas rates exceeding $2/\lambda_{\max}$ lead to divergence. More recently, Lewkowycz et al. (2020) identified a "catapult" learning regime characterized by $2/\lambda_{\max} \leq \eta^* \leq 12/\lambda_{\max}$, in which modern architectures achieve optimal performance. More research has further explored the intricacies of the interplay between $\lambda_{\max}$ and $\eta^*$ with a consensus that these two quantities exhibit an inversely proportional relationship (Pan et al., 2021; Cohen et al., 2021; Kalra & Barkeshli, 2024).

### 5.2. Sharpness Analysis in LoRA

For our LoRA fine-tuning problem, we leverage the downstream dataset to compute the Hessian matrix of the loss function and focus exclusively on the trainable LoRA parameters (Yang et al., 2023; Zhao et al., 2024b; Yu et al., 2025b). Instead of concatenating LoRA parameters across all layers, we follow standard practices in LLMs to estimate $\lambda_{\max}$ in a block-wise manner (Zhang et al., 2024b; Wang et al., 2025a; Ilin, 2025) at the initialization point. Formally, we calculate the layer-wise metric as $\lambda_{\max}^l = \lambda_{\max}(\mathbf{H}^l)$, where $\mathbf{H}^l$ represents the Hessian corresponding to parameters $\theta^l = \{B_0^l, A_0^l\}$, with $l$ indexing through matrix types and Transformer layers. The Lanczos algorithm (Lanczos, 1950) and Hessian-vector Products are utilized to estimate the top eigenvalue without explicitly forming $\mathbf{H}$. Implementation details are provided in Appendix G.1. While DoRA shares LoRA's initialization, implying identical initial Hessians, its unique architectural design may lead to distinct Hessian evolution throughout training, rendering the analysis non-trivial; we thus defer the investigation of DoRA to future work.

Specifically, let the Hessian for the Query projection matrix in the $i$-th layer be $\mathbf{H}_t^{Q,i}$ for $t =$PiSSA, MiLoRA, Init[AB], and LoRA, we further denote their corresponding maximum eigenvalues by $\lambda_{\max,t}^{Q,i}$. Then, we normalize the maximum eigenvalues from PiSSA, MiLoRA, and Init[AB] by that from LoRA, and plot the distribution across layers in Figure 6, i.e. $\lambda_{\max,t}^{Q,i}/\lambda_{\max,\text{LoRA}}^{Q,i}$ for $t =$PiSSA, MiLoRA, Init[AB] and $i = 1, \ldots, L$. The results reveal that PiSSA initializes trainable parameters in a state of significantly higher curvature compared to other methods. This observation corroborates our prior learning rate tuning experiments, explaining why PiSSA requires a much lower learning rate. For other initialization variants, however, the eigenvalue magnitudes become more similar to those of vanilla LoRA. Notably on Qwen, Init[AB] and MiLoRA exhibit slightly larger $\lambda_{\max}$ ($\approx 2\times$) compared to LoRA, supporting their lower optimal learning rates by a factor of $1.8\times$ and $3.2\times$ in Figure 1, respectively.

## 6. Conclusion

Motivated by the increasing number of LoRA variants and the insufficient hyperparameter tuning in many studies, in this work, we conducted a systematic re-evaluation of five LoRA PEFT methods under a unified evaluation protocol. Based on the comprehensive hyperparameter experiments, we conclude that **improper learning rate gives a false sense of LoRA advancements**. In our studies, we also pointed out the scenarios where one might be in favor of a specific variant that, albeit likely to produce generally comparable performance, marginally outperforms other variants. It is worth noting that these improvements often lack universality, with vanilla LoRA frequently matching or even outperforming them. By elucidating the disparate optimal learning rate ranges through Hessian analysis, we hope our study encourages future PEFT research to adopt a more comprehensive hyperparameter search protocol, ensuring reliable advancements in the field.

## Impact Statement

The objective of this paper is to advance Machine Learning research. Although there may be potential societal consequences of our work, we do not identify any specific ones that must be specifically highlighted here.

We acknowledge that this paper is subject to several limitations, primarily due to computational constraints. Specifically, we focused our investigation on decoder-only LLMs up to the 7B parameter scale, restricting our evaluation to mathematical and code generation tasks. Consequently, the scalability of our findings to larger foundation models and their generalization to diverse downstream tasks remain to be verified. We also caution that our results may not extend to untested model architectures or other advanced LoRA variants. We refer the readers to Appendix Sec. H for more details.

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

# Contents

# A. Comprehensive Study of Hyperparameters in Prior Work

## A.1. Survey Criteria

To generate the statistics presented in Figure 2, we curated a dataset comprising 52 papers, consisting of 42 studies published at major AI conferences or journals, 6 high-impact arXiv preprints (exceeding 40 citations), and 4 recent preprints released within the last six months. For each paper, we examined whether the authors reported performance metrics under learning rate or batch size tuning, and whether comparisons across different ranks were provided. The selection criteria for inclusion were as follows:

1. The primary objective of the proposed method is to enhance fine-tuning effectiveness (i.e., aiming for higher accuracy with equivalent parameter counts or sustained performance with greater parameter efficiency).

2. Vanilla LoRA is explicitly employed as a baseline for performance comparison.

In assessing hyperparameter tuning, **our analysis focuses exclusively on decoder-only LLMs**, excluding encoder-only (Devlin et al., 2019), encoder-decoder (Raffel et al., 2020) architectures, Vision Transformers (Dosovitskiy, 2020), and Vision-Language Models (Alayrac et al., 2022), as these lie outside the scope of this work. Consequently, papers lacking experiments on decoder-only LLMs are excluded from our statistics (e.g., Zhang et al. (2023b); Liu et al. (2023b); Xu et al. (2025); Zhang et al. (2025d); Edalati et al. (2025); Yuan et al. (2024)). Moreover, we exclude papers focusing on objectives other than PEFT efficiency, such as parameter-efficient pretraining (Lialin et al., 2023), continual learning (Wang et al., 2023; Zhao et al., 2024a), and quantization (Dettmers et al., 2023; Xu et al., 2023).

Given that some studies may tune hyperparameters only for their proposed methods while leaving baselines untuned (e.g., by adopting settings from prior work without modification), **we rigorously verified whether the vanilla LoRA baseline underwent tuning**. Specifically, we consider the learning rate and batch size to be "tuned" only if they are evaluated across at least three distinct values. Consequently, studies such as MiLoRA (which compared two sets of hyperparameter setups) or LoRA-GA (Wang et al., 2024c) (which tested learning rate in {1e-5, 5e-5}) do not qualify as tuning under our criteria. For rank, we require comparisons across at least two distinct values. Crucially, if a study varies the rank for its proposed method but benchmarks against a fixed-rank vanilla LoRA, we do not classify the baseline rank as tuned (e.g., Bałazy et al. (2024); Yuan et al. (2024); Li et al. (2024)).

During the data curation process, we observed that verifying specific hyperparameter tuning details can be non-trivial in some cases. The difficulty arises from discrepancies between paper versions (e.g., ablation studies on hyperparameters added to the Appendix post-publication), incomplete descriptions of experimental setups, or underspecified hyperparameter settings. Additionally, we frequently observed papers performing hyperparameter tuning only on smaller encoder-only LLMs (e.g., RoBERTa (Liu et al., 2019)). In strict adherence to our inclusion criteria, we do not categorize these instances as "tuned". While these ambiguities complicate binary categorization, we have made every effort to ensure accuracy. We emphasize that this list is curated solely to present the current state of hyperparameter tuning practices in LoRA PEFT research, and we welcome future contributions or corrections to further refine this collection.

## A.2. Comprehensive List of Papers

A comprehensive list of the reviewed papers is detailed in Table 2. Specifically, "arXiv Date" marks the release date of the first version on arXiv (denoted as "–" if unavailable). "Pub. Date" refers to the formal publication date of the venue, where "–" indicates the paper has not yet been formally published. Note that the table is sorted primarily by Pub. Date, followed by arXiv Date.

*Table 2.* Publication dates, venues, and experimental configurations in prior LoRA PEFT studies. The table summarizes decoder-only LLMs and tasks, noting whether the vanilla LoRA baseline involved learning rate (**LR**) or batch size (**BS**) tuning and offered comparisons across different **Ranks**. A positive entry (✓) indicates the configuration was provided for at least one model-task combination; ✗ denotes otherwise. The symbol ∗ denotes a workshop paper.

| Method | arXiv Date | Pub. Date | Venue | Decoder-only LLM | Fine-tuned Task | LR | BS | Rank |
|---|---|---|---|---|---|---|---|---|
| **DyLoRA** (2023) | 2022-10 | 2023-05 | EACL | GPT-2 Medium | NLG | ✗ | ✗ | ✓ |
| **GLoRA** (2023) | 2023-06 | – | arXiv | Llama-1-7B
Llama-2-7B | NLG | ✗ | ✗ | ✗ |

| Method | arXiv Date | Pub. Date | Venue | Decoder-only LLM | Fine-tuned Task | LR | BS | Rank |
|---|---|---|---|---|---|---|---|---|
| **LoRA-FA** (2023a) | 2023-08 | – | arXiv | Llama-1-7B
Llama-2-7B | Commonsense | ✗ | ✗ | ✗ |
| **Laplace-LoRA** (2023) | 2023-08 | 2024-05 | ICLR | Llama-1-7B
Llama-2-7B | Commonsense | ✗ | ✗ | ✗ |
| **VeRA** (2023a) | 2023-10 | 2024-05 | ICLR | GPT-2 Medium/Large
Llama-1-7B/13B
Llama-2-7B/13B | NLG
Instruction Following | ✓ | ✗ | ✓ |
| **BOFT** (2023b) | 2023-11 | 2024-05 | ICLR | Llama-2-7B | Instruction Following
Math | ✗ | ✗ | ✓ |
| **MoRA** (2024) | 2024-05 | – | arXiv | Llama-2-7B/13B | UUID
Math
Instruction Following | ✓ | ✗ | ✓ |
| **Delta-LoRA** (2023) | 2023-09 | – | arXiv | GPT-2 Medium | NLG | ✗ | ✗ | ✗ |
| **Tied-LoRA** (2024) | 2023-11 | 2024-06 | NAACL | GPT-2B-001
Llama-2-7B | NLG
Commonsense
Math | ✗ | ✗ | ✓ |
| **LoRETTA** (2024b) | 2024-02 | 2024-06 | NAACL | Llama-2-7B/13B/70B | NLG
GLUE | ✗ | ✗ | ✗ |
| **AutoLoRA** (2024a) | 2024-03 | 2024-06 | NAACL | GPT-2 Medium | NLG | ✗ | ✗ | ✗ |
| **ALoRA** (2024b) | 2024-05 | 2024-06 | NAACL | GPT2-Large
Llama-2-7B | NLG
GLUE
Instruction Following | ✗ | ✗ | ✓ |
| **RoSA** (2024) | 2024-01 | 2024-07 | ICML | Llama-2-7B | NLG
Math
Code
Instruction Following | ✓ | ✗ | ✗ |
| **LoRA+** (2024b) | 2024-02 | 2024-07 | ICML | GPT-2
Llama-1-7B | GLUE
Instruction Following | ✓ | ✗ | ✗ |
| **scaled AdamW** (2024) | 2024-02 | 2024-07 | ICML | GPT-2 Medium
Mistral-7B-V0.1 | NLG
GLUE | ✓ | ✗ | ✓ |
| **DoRA** (2024a) | 2024-02 | 2024-07 | ICML | Llama-1-7B/13B
Llama-2-7B
Llama-3-8B | Commonsense | ✗ | ✗ | ✓ |
| **FLORA** (2024) | 2024-02 | 2024-07 | ICML | GPT-2 -base/XL | Summarization
Translation | ✓ | ✗ | ✓ |
| **FourierFT** (2024) | 2024-05 | 2024-07 | ICML | GPT-2 Medium/Large
Llama-1-7B/13B
Llama2-7B/13B | NLG
Instruction Following | ✓ | ✓ | ✓ |
| **ResLoRA** (2024) | 2024-02 | 2024-08 | ACL | Llama-2-7B | Math
Commonsense | ✗ | ✗ | ✓ |
| **PLoRA** (2024b) | 2024-02 | – | arXiv | Llama-1-7B | Instruction Following
Math | ✓ | ✗ | ✓ |
| **OLoRA** (2024) | 2024-06 | – | arXiv | Mistral-7B
LLaMA-2-7B
Tiny Llama-1.1B
Gemma-2B
OPT-1.3B | Commonsense
Instruction Following | ✗ | ✗ | ✓ |
| **LamDA** (2024) | 2024-06 | 2024-11 | EMNLP | Llama-2-7B | NLG
Math
Commonsense | ✗ | ✗ | ✓ |

| Method | arXiv Date | Pub. Date | Venue | Decoder-only LLM | Fine-tuned Task | LR | BS | Rank |
|---|---|---|---|---|---|---|---|---|
| **LoRA-drop** (2025) | 2024-02 | 2025-01 | COLING | Llama-2-7b | NLG
Summarization
GLUE
Math | ✗ | ✗ | ✗ |
| **PiSSA** (2024a) | 2024-04 | 2024-12 | NeurIPS | Llama-2-7B / 13B
Llama-3-8B / 70B
Mistral-7B-v0.1
Gemma-7B
Qwen1.5-7B
Yi-1.5-34B
DeepSeek-MoE-16B
Mixtral-8x7B | Math
Code
Instruction Following | ✗ | ✗ | ✓ |
| **VB-LoRA** (2024) | 2024-05 | 2024-12 | NeurIPS | GPT-2 Medium/Large
Llama-2-7B/13B
Mistral-7B-v0.1/Gemma-7B | NLG
Math
Instruction Following | ✗ | ✗ | ✗ |
| **HRA** (2024) | 2024-05 | 2024-12 | NeurIPS | Llama-2-7B | Math | ✗ | ✗ | ✗ |
| **CorDA** (2024a) | 2024-06 | 2024-12 | NeurIPS | Llama-2-7B/13B
Llama-3-8B
Gemma-2-9B | Math
Code
Instruction Following
World Knowledge | ✗ | ✗ | ✓ |
| **LoRA-GA** (2024c) | 2024-07 | 2024-12 | NeurIPS | Llama-2-7B | Math
Code
Instruction Following | ✗ | ✗ | ✗ |
| **RoAd** (2024) | 2024-09 | 2024-12 | NeurIPS | Llama-1-7B/13B
Llama-2-7B
Llama-3-8B | Math
Commonsense | ✗ | ✗ | ✓ |
| **AG-LoRA** (2025) | – | 2025-01 | IEEE Access | Llama-1-7B | Commonsense Reasoning | ✗ | ✗ | ✓ |
| **LoRA-Dash** (2024) | 2024-09 | 2025-04 | ICLR | Llama-1-7B
Llama-2-7B
Llama-3-8B
Qwen2.5-7B | GLUE
Commonsense | ✗ | ✗ | ✓ |
| **KaSA** (2024a) | 2024-09 | 2025-04 | ICLR | Llama-1-7B
Llama-2-7B
Llama-3-8B
Qwen2.5-7B | GLUE
Commonsense
Instruction Following | ✗ | ✗ | ✓ |
| **RandLoRA** (2025) | 2025-02 | 2025-04 | ICLR | GPT-2 Medium
Qwen2-0.5B
Phi3-3B
Llama3-8B | NLG
Commonsense | ✗ | ✗ | ✓ |
| **DeLoRA** (2025) | 2025-03 | 2025-04 | ICLR | Llama-2-7B
Llama3-8B | Commonsense | ✓ | ✗ | ✓ |
| **HiRA** (2025) | – | 2025-04 | ICLR | Llama-2-7B
Llama-3-8B | Math
Commonsense
Dialogue Generation | ✗ | ✗ | ✓ |
| **MiLoRA** (2024b) | 2024-06 | 2025-04 | NAACL | Llama-2-7B
Llama-3-8B
Qwen2.5-7B | Math
Commonsense
Instruction Following | ✗ | ✗ | ✗ |
| **SSMLoRA** (2025a) | 2025-02 | 2025-04 | NAACL | GPT-2
Llama-2-7B/13B | GLUE | ✓ | ✗ | ✓ |
| **MiSS** (2025) | 2024-09 | 2025-07 | ICML∗ | Llama2-7B/13B
Mistral-7B
Qwen3-4B
Llama-3.2-3B | Math
Code
Instruction Following | ✗ | ✗ | ✓ |

| Method | arXiv Date | Pub. Date | Venue | Decoder-only LLM | Fine-tuned Task | LR | BS | Rank |
|---|---|---|---|---|---|---|---|---|
| **LoRA-One** (2025g) | 2025-02 | 2025-07 | ICML | Llama-2-7B | Math
Code
Instruction Following | ✓ | ✓ | ✗ |
| **Init[AB]** (2025) | 2025-05 | 2025-07 | ICML | Llama-3-8B | Arithmetic
Commonsense | ✓ | ✗ | ✗ |
| **C3A** (2025) | 2024-07 | 2025-07 | ACL | Llama-2-7B
Llama-3-8B/70B
Mistral-7B
Mistral-8x7B | Math
Code
Commonsense | ✗ | ✗ | ✗ |
| **SuLoRA** (2025) | – | 2025-07 | ACL | Llama-2-7B | Instruction Following | ✗ | ✗ | ✓ |
| **BiDoRA** (2025) | 2024-10 | 2025-08 | TMLR | GPT-2 Medium | NLG | ✗ | ✗ | ✗ |
| **HD-PiSSA** (2025c) | 2025-05 | 2025-11 | EMNLP | Llama-2-7B
Llama-3-8B
Mistral-7b-v0.1 | Math
Code | ✗ | ✗ | ✓ |
| **LoSiA** (2025b) | 2025-07 | 2025-11 | EMNLP | Gemma 2B
Llama-2-7B/13B | Math
Code
Commonsense
Instruction Folllowing | ✓ | ✗ | ✓ |
| **Sensitivity-LoRA** (2025b) | 2025-09 | 2025-11 | EMNLP | GPT-2 Large
Qwen2.5-7B/32B
Llama-3.1-8B | NLG
Instruction Following | ✗ | ✗ | ✗ |
| **OHoRA** (2025f) | – | 2025-11 | EMNLP | Llama-2-7B
Llama-3-8B
Gemma-7B
Llama-3.1-8B-Inst | Math
Code
Commonsense
Instruction Following | ✗ | ✗ | ✓ |
| **AuroRA** (2025) | 2025-05 | 2025-12 | NeurIPS | Llama-3-8B | Commonsense | ✗ | ✗ | ✓ |
| **DropLoRA** (2025) | 2025-08 | – | arXiv | Llama-2-7B
Llama-3-8B | Math
Code
Commonsense
Instruction Following | ✗ | ✗ | ✓ |
| **PrunedLoRA** (2025b) | 2025-09 | – | arXiv | Llama-3-8B | | ✓ | ✗ | ✓ |
| **LoRA-DA** (2025e) | 2025-10 | – | arXiv | Llama-2-7B | Math
Commonsense | ✗ | ✗ | ✓ |
| **ABM-LoRA** (2025) | 2025-11 | – | arXiv | Llama-2-7B | Instruction Following | ✗ | ✗ | ✓ |

# B. Fine-tuning Implementation Details

## B.1. Models

Following standard practice in PEFT research to ensure results purely reflect the training data, we use base versions (not instruction-tuned) for all models (Hu et al., 2022; Meng et al., 2024a; Wang et al., 2024b; Li et al., 2025; Liu et al., 2024a). Specifically, we utilize the official checkpoints hosted on Hugging Face: `Qwen3-0.6B-Base`[1], `gemma-3-1b-pt`[2], and `Llama-2-7b-hf`[3].

## B.2. Hyperparameter Search Ranges

See Table 3 for hyperparameter search ranges for each model-task combination. For all experiments on Qwen and Gemma, we conduct three independent trainings and report the mean and standard deviation.

| Model | Task | Rank ($r$) | Batch ($B$) | Learning Rate ($\eta$) |
|---|---|---|---|---|
| **Qwen3-0.6B** | Math | 8 | {16, 64, 128} | 1.1247e-5 – 6.3246e-3 |
| | | 128 | 64 | 2.0000e-6 – 2.0000e-3 |
| **Gemma-3-1B** | Math | {4, 8, 16, 32, 64, 128, 256} | 64 | 1.1247e-5 – 6.3246e-3 |
| | | 128 | {16, 64, 128} | 1.1247e-5 – 6.3246e-3 |
| | Code | {4, 8, 16, 32, 64, 128, 256} | 64 | 1.1247e-5 – 6.3246e-3 |
| **Llama-2-7B** | Math | {8, 32, 128} | {16, 128} | 2.0000e-5 – 3.5566e-3 |
| | Code | {8, 32, 128} | {16, 128} | 2.0000e-5 – 3.5566e-3 |

*Table 3.* Summary of models, tasks, ranks, and hyperparameter search ranges. Learning rates are tuned evenly in logarithmic scale: {1.1247e*, 2.0000e*, 3.5566e*, 6.3246e*} per order of magnitude.

## B.3. Fixed Training Hyperparameters

Except for tunable hyperparameters (i.e., learning rate and batch size), all other training configurations remain fixed and the same for all experiments; the values are summarized in Table 4. Note that these configurations primarily follow PiSSA, thus may differ from those of other considered PEFT methods. For example, MiLoRA, DoRA, and Init[AB] employ linear decay instead of cosine annealing for learning rate scheduling. Additionally, MiLoRA and DoRA use fixed warmup steps (100) rather than 3% of total training steps, and apply a dropout rate of 0.05 instead of no dropout. Furthermore, while we place low-rank adapters on all linear layers, these methods exclude output projections (out_proj) or gate matrices (gate_proj) in several of their experiments.

| Configuration | Value |
|---|---|
| Epoch | 1 |
| Scaling factor ($\alpha$) | $r$ |
| Optimizer | AdamW (Loshchilov & Hutter, 2017) |
| LR scheduler | Cosine annealing with warmup |
| Warmup ratio | 3% |
| Dropout | None |
| Weight Decay | None |
| Adapter placement | All linear layers |
| Base model precision | BFloat16 (Wang & Kanwar, 2019)[4] |
| Adapter precision | Float32 |
| Max sequence length | 512 |

*Table 4.* Fixed training configurations across all experiments. Scaling factor $\alpha$ equals the LoRA rank $r$. Adapters are applied to all linear projection layers (q_proj, k_proj, v_proj, o_proj, gate_proj, up_proj, down_proj).

---

[1] https://huggingface.co/Qwen/Qwen3-0.6B-Base

[2] https://huggingface.co/google/gemma-3-1b-pt

[3] https://huggingface.co/meta-llama/Llama-2-7b-hf

[4] Following PiSSA's codebase, normalization layers and gate_proj matrices of the pretrained model are converted back to Float32 after the BFloat16 sampling and before training.

## B.4. Data, Code, Libraries, and Hardware

The codebase of Meng et al. (2024a)[5] is used to run LoRA, PiSSA, and DoRA (with DoRA implemented via the PEFT package (Mangrulkar et al., 2022)), and we further extend this codebase to support other PEFT methods, including MiLoRA[6] and Init[AB][7]. Note that while the original LoRA paper used Kaiming Normal initialization, we follow its official implementation and the widely-used PEFT library to use Kaiming Uniform instead in our experiments. The results are expected to be similar (cf. Meng et al. (2024a, Table 2)). Additionally, while Li et al. (2025) also proposed a variant, Init[AB+], which does not require $W_{\text{res}}$ and shows no discernible performance difference, we chose to implement the default Init[AB]. For the training data, we directly utilized the preprocessed dataset released by Meng et al. (2024a)[8].

PyTorch (Paszke et al., 2019) version 2.7.1 is used for implementation. All experiments are conducted on four GPUs (either $4\times$ Nvidia RTX 3090 or $4\times$ Nvidia A6000). We employ DeepSpeed (Rasley et al., 2020) for parallel training and vLLM (Kwon et al., 2023) for parallel inference. During inference, we apply greedy decoding (i.e., temperature set to 0), and utilize the EvalPlus (Liu et al., 2023a) framework to evaluate pass@1 for code generation tasks.

All fine-tuning experiments (except those for Llama-2-7B) are conducted with three independent runs and reported with means and standard deviations. The sources of randomness are controlled by explicitly fixing random seeds across Python, NumPy, and PyTorch using the code snippet shown below.

```
def seed_everything(seed):
    random.seed(seed)
    os.environ['PYTHONHASHSEED'] = str(seed)
    np.random.seed(seed)
    torch.manual_seed(seed)
    torch.cuda.manual_seed(seed)
    torch.backends.cudnn.deterministic = True
    torch.backends.cudnn.benchmark = True
```

# C. On LoRA Scaling Factor

The configuration of the LoRA alpha parameter ($\alpha$) generally follows two paradigms: (1) setting it to a fixed constant across LoRA ranks (typically 32 or 64), or (2) scaling it with the LoRA rank, often following $\alpha = r$ or $\alpha = 2r$, which results in a scaling factor $\gamma_r$ of 1 or 2, respectively. The configurations adopted for decoder-LLMs in the considered LoRA variants are summarized as follows:

- **PiSSA:** $\gamma_r = 1$ ($\alpha = r$ for all $r$).

- **MiLoRA:** $\gamma_r = 2$ for vanilla LoRA; $\gamma_r = 1$ for both MiLoRA and PiSSA.

- **Init[AB]:** $\gamma_r = 1$ ($r = 16$ and $\alpha = 16$).

- **DoRA:** $\gamma_r = 2$ ($\alpha = 2r$ for all $r$).

Evidently, the methods considered in this paper adhere to the second paradigm. Consequently, we adopt the setting $\alpha = r$ ($\gamma_r = 1$) for all our experiments. We refer readers to several prior studies that have explored the optimal LoRA scaling factor: Kalajdzievski (2023) argued that $\alpha$ should scale with the square root of $r$, rather than linearly ($\alpha \propto r$), though the optimal $\alpha$ setup remains unclear. Empirically, Biderman et al. (2024) demonstrated through joint sweeps of $\alpha$ and learning rates that $\alpha = 2r$ is the optimal choice, with $\alpha = r$ performing only marginally worse (cf. Biderman et al. (2024, Appendix B.2, Figure S3)). Notably, Zhang et al. (2025h) recently unified the learning rate, scaling factor, and initialization under a single theoretical framework, suggesting that tuning the learning rate is theoretically equivalent to tuning the scaling factor (Fan et al., 2025). This further validates our decision to fix the scaling factor in our experiments.

---

[5]https://github.com/GraphPKU/PiSSA
[6]https://github.com/sufenlp/MiLoRA
[7]https://github.com/Leopold1423/non_zero_lora-icml25
[8]https://huggingface.co/datasets/fxmeng/pissa-dataset

# D. Additional Experiments

## D.1. Varying Training Duration

In this section, we examine whether different methods consistently peak at comparable performance levels under varying training durations. Specifically, we vary the training duration by scaling the number of MetaMath training samples from 5k up to the full 395k (Figure 7).

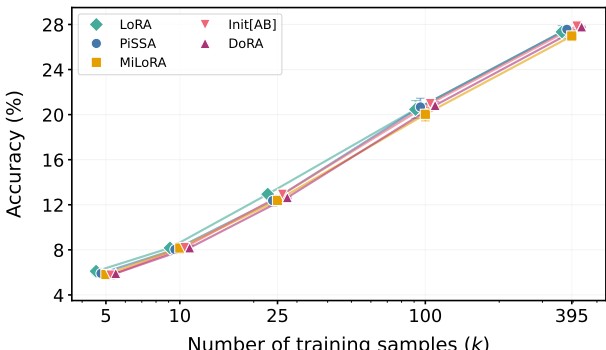

*Figure 7.* Best achievable performance of LoRA and its variants across different training sample sizes on mathematical reasoning with Gemma-3-1B ($r = 128$, $B = 64$). Once the learning rate is properly tuned, all methods exhibit nearly identical improvement trends as the number of training samples increases. Results are reported with mean and standard deviation over three runs.

## D.2. Varying Adapter Ranks on Llama

In Sec. 4.3.2, we analyzed the behavior of LoRA PEFT methods as the adapter rank varies on Gemma across math and code tasks. Here, we present a corresponding analysis on Llama in Figure 8, reporting the best performance achieved under joint optimization of learning rate and batch size ($B \in \{16, 128\}$). Similar trends can be observed, where all methods exhibit comparable performance improvement trends as the rank increases. Although under specific tasks or rank settings, one might favor a particular variant that marginally outperforms others, it is worth noting again that these improvements often lack universality, with vanilla LoRA frequently matching or even outperforming them.

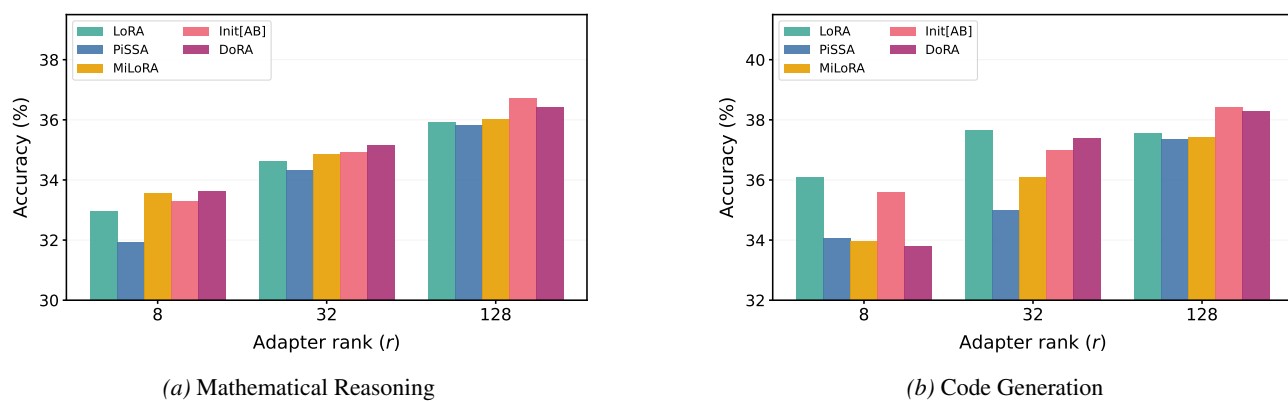

*(a)* Mathematical Reasoning          *(b)* Code Generation

*Figure 8.* Best achievable performance of LoRA and its advanced variants across adapter ranks on Llama-2-7B ($B \in \{16, 128\}$).

# E. Details of Hyperparameter Search Results

Sections E.1, E.2, and E.3 present detailed hyperparameter search results on Qwen, Gemma, and Llama, respectively. Note that Tables 6, 10, and 13 correspond to the detailed numerical results for Figures 1, 4a, and 4b in the main text, respectively. Additionally, Tables 8, 9, and 10 provide the hyperparameter search details across adapter ranks for Appendix Figure 8a, while Tables 11, 12, and 13 correspond to Appendix Figure 8b.

## E.1. Qwen3-0.6B

### E.1.1. MATHEMATICAL REASONING

| Methods | Batch Size | Learning Rate | | | | | | | | | | | |
|---|---|---|---|---|---|---|---|---|---|---|---|---|---|
| | | 1.12e-05 | 2.00e-05 | 3.56e-05 | 6.32e-05 | 1.12e-04 | 2.00e-04 | 3.56e-04 | 6.32e-04 | 1.12e-03 | 2.00e-03 | 3.56e-03 | 6.32e-03 |
| LoRA | 16 | $38.94_{\pm1.02}$ | $46.49_{\pm1.17}$ | $47.78_{\pm0.47}$ | $47.85_{\pm0.42}$ | $48.13_{\pm0.40}$ | $48.90_{\pm0.41}$ | $48.41_{\pm0.59}$ | $\mathbf{49.05}_{\pm0.49}$ | $47.64_{\pm0.86}$ | $44.00_{\pm0.45}$ | $26.59_{\pm21.82}$ | $5.03_{\pm5.33}$ |
| | 64 | $29.53_{\pm0.18}$ | $33.02_{\pm0.28}$ | $39.15_{\pm0.33}$ | $46.49_{\pm0.25}$ | $48.16_{\pm0.24}$ | $48.39_{\pm0.29}$ | $48.95_{\pm0.23}$ | $\mathbf{48.99}_{\pm0.40}$ | $48.73_{\pm0.10}$ | $48.14_{\pm0.42}$ | $43.92_{\pm0.48}$ | $1.28_{\pm0.05}$ |
| | 128 | $22.51_{\pm0.00}$ | $30.22_{\pm1.63}$ | $33.55_{\pm0.00}$ | $40.64_{\pm1.48}$ | $48.31_{\pm0.00}$ | $48.38_{\pm0.19}$ | $48.38_{\pm0.00}$ | $48.69_{\pm0.57}$ | $\mathbf{48.75}_{\pm0.00}$ | $48.33_{\pm0.54}$ | $1.06_{\pm0.00}$ | $0.71_{\pm0.00}$ |
| DoRA | 16 | $42.03_{\pm1.63}$ | $47.36_{\pm0.59}$ | $48.10_{\pm0.12}$ | $48.10_{\pm0.48}$ | $48.29_{\pm0.08}$ | $\mathbf{48.80}_{\pm0.64}$ | $48.60_{\pm0.17}$ | $48.70_{\pm0.05}$ | $46.30_{\pm0.29}$ | $42.67_{\pm0.19}$ | $35.93_{\pm0.44}$ | $1.31_{\pm0.19}$ |
| | 64 | $38.69_{\pm1.24}$ | $37.60_{\pm1.22}$ | $40.65_{\pm1.58}$ | $47.06_{\pm0.50}$ | $48.41_{\pm0.31}$ | $48.03_{\pm0.18}$ | $\mathbf{49.07}_{\pm0.03}$ | $48.87_{\pm0.82}$ | $48.55_{\pm0.43}$ | $47.31_{\pm0.48}$ | $44.61_{\pm0.00}$ | $0.89_{\pm0.00}$ |
| | 128 | $32.27_{\pm0.00}$ | $36.85_{\pm0.98}$ | $36.85_{\pm0.00}$ | $43.03_{\pm1.11}$ | $48.08_{\pm0.00}$ | $47.94_{\pm0.30}$ | $48.13_{\pm0.00}$ | $48.41_{\pm0.72}$ | $\mathbf{48.52}_{\pm0.00}$ | $47.75_{\pm0.47}$ | $46.15_{\pm0.00}$ | $42.11_{\pm0.00}$ |
| Init[AB] | 16 | $36.53_{\pm2.20}$ | $41.67_{\pm1.99}$ | $45.47_{\pm1.30}$ | $48.07_{\pm0.70}$ | $48.28_{\pm0.72}$ | $\mathbf{48.66}_{\pm0.31}$ | $48.53_{\pm0.48}$ | $48.18_{\pm0.49}$ | $46.79_{\pm0.19}$ | $42.32_{\pm0.39}$ | $38.65_{\pm0.80}$ | $20.67_{\pm27.96}$ |
| | 64 | $35.78_{\pm0.54}$ | $35.15_{\pm1.09}$ | $37.85_{\pm0.08}$ | $40.04_{\pm1.50}$ | $45.03_{\pm1.13}$ | $48.34_{\pm0.29}$ | $48.53_{\pm0.07}$ | $48.45_{\pm0.50}$ | $\mathbf{48.68}_{\pm0.15}$ | $47.11_{\pm0.43}$ | $43.13_{\pm0.80}$ | $1.36_{\pm0.06}$ |
| | 128 | $32.27_{\pm0.00}$ | $32.44_{\pm0.70}$ | $36.44_{\pm0.00}$ | $35.29_{\pm1.90}$ | $42.67_{\pm0.00}$ | $47.06_{\pm1.29}$ | $\mathbf{48.81}_{\pm0.00}$ | $48.57_{\pm0.39}$ | $48.13_{\pm0.00}$ | $48.34_{\pm0.66}$ | $46.95_{\pm0.00}$ | $0.62_{\pm0.00}$ |
| MiLoRA | 16 | $39.42_{\pm0.87}$ | $45.09_{\pm0.19}$ | $44.76_{\pm0.78}$ | $45.92_{\pm0.81}$ | $49.16_{\pm0.37}$ | $\mathbf{49.36}_{\pm0.09}$ | $48.93_{\pm0.37}$ | $48.08_{\pm0.06}$ | $46.09_{\pm0.55}$ | $43.39_{\pm0.52}$ | $25.53_{\pm20.93}$ | $1.47_{\pm0.07}$ |
| | 64 | $32.25_{\pm1.20}$ | $38.33_{\pm1.24}$ | $45.72_{\pm0.88}$ | $44.08_{\pm1.50}$ | $47.32_{\pm0.05}$ | $48.69_{\pm0.32}$ | $\mathbf{49.40}_{\pm0.01}$ | $49.06_{\pm0.18}$ | $48.90_{\pm0.37}$ | $47.02_{\pm0.38}$ | $43.98_{\pm1.06}$ | $1.07_{\pm0.50}$ |
| | 128 | $30.83_{\pm0.00}$ | $33.71_{\pm0.34}$ | $35.50_{\pm0.00}$ | $40.49_{\pm0.39}$ | $44.12_{\pm0.00}$ | $48.08_{\pm0.42}$ | $48.78_{\pm0.00}$ | $\mathbf{49.37}_{\pm0.19}$ | $48.96_{\pm0.00}$ | $48.12_{\pm0.74}$ | $46.95_{\pm0.00}$ | $1.47_{\pm0.00}$ |
| PiSSA | 16 | $47.10_{\pm0.28}$ | $44.80_{\pm0.71}$ | $46.45_{\pm0.61}$ | $\mathbf{48.37}_{\pm0.36}$ | $48.30_{\pm0.26}$ | $47.42_{\pm0.22}$ | $45.47_{\pm0.36}$ | $42.43_{\pm0.03}$ | $38.83_{\pm0.40}$ | $33.14_{\pm0.41}$ | $24.63_{\pm0.98}$ | $14.82_{\pm19.10}$ |
| | 64 | $44.20_{\pm0.12}$ | $44.12_{\pm0.69}$ | $47.54_{\pm0.50}$ | $48.25_{\pm0.14}$ | $48.17_{\pm0.45}$ | $\mathbf{48.43}_{\pm0.12}$ | $47.94_{\pm0.39}$ | $47.23_{\pm0.19}$ | $43.94_{\pm0.04}$ | $40.15_{\pm0.20}$ | $35.37_{\pm0.55}$ | $14.66_{\pm19.34}$ |
| | 128 | $41.95_{\pm0.00}$ | $43.64_{\pm0.51}$ | $43.97_{\pm0.00}$ | $46.42_{\pm0.10}$ | $48.18_{\pm0.00}$ | $\mathbf{48.24}_{\pm0.18}$ | $47.41_{\pm0.00}$ | $47.70_{\pm0.36}$ | $45.66_{\pm0.00}$ | $43.54_{\pm0.41}$ | $38.98_{\pm0.00}$ | $33.05_{\pm0.00}$ |

*Table 5.* Performance of **Qwen3-0.6B** fine-tuned on **mathematical reasoning** tasks with **rank = 8**.

| Methods | Batch Size | Learning Rate | | | | | | | | | | | |
|---|---|---|---|---|---|---|---|---|---|---|---|---|---|
| | | 2.00e-06 | 3.56e-06 | 6.32e-06 | 1.12e-05 | 2.00e-05 | 3.56e-05 | 6.32e-05 | 1.12e-04 | 2.00e-04 | 3.56e-04 | 6.32e-04 | 1.12e-03 | 2.00e-03 |
| LoRA | 64 | $21.48_{\pm0.53}$ | $24.52_{\pm0.44}$ | $31.94_{\pm0.58}$ | $43.07_{\pm1.16}$ | $48.37_{\pm0.19}$ | $48.81_{\pm0.14}$ | $49.27_{\pm0.34}$ | $49.46_{\pm0.56}$ | $\mathbf{49.60}_{\pm0.18}$ | $48.95_{\pm0.20}$ | $47.08_{\pm0.22}$ | $40.76_{\pm0.96}$ | $1.27_{\pm0.15}$ |
| DoRA | 64 | $36.84_{\pm0.86}$ | $38.19_{\pm1.74}$ | $39.29_{\pm0.36}$ | $44.72_{\pm0.96}$ | $48.09_{\pm0.74}$ | $49.01_{\pm0.44}$ | $49.25_{\pm0.33}$ | $\mathbf{49.45}_{\pm0.25}$ | $49.33_{\pm0.45}$ | $49.32_{\pm0.71}$ | $46.92_{\pm0.28}$ | $40.24_{\pm0.62}$ | $10.14_{\pm15.64}$ |
| Init[AB] | 64 | $33.40_{\pm2.16}$ | $35.81_{\pm1.72}$ | $41.20_{\pm1.83}$ | $47.27_{\pm0.89}$ | $49.27_{\pm0.13}$ | $49.02_{\pm0.11}$ | $48.81_{\pm0.17}$ | $\mathbf{49.29}_{\pm0.23}$ | $48.51_{\pm0.44}$ | $47.37_{\pm0.39}$ | $44.81_{\pm0.34}$ | $39.41_{\pm0.49}$ | $0.93_{\pm0.41}$ |
| MiLoRA | 64 | $33.62_{\pm0.39}$ | $38.35_{\pm0.46}$ | $43.37_{\pm0.41}$ | $48.30_{\pm0.27}$ | $49.08_{\pm0.23}$ | $48.74_{\pm0.32}$ | $\mathbf{49.17}_{\pm0.38}$ | $49.08_{\pm0.14}$ | $48.22_{\pm0.20}$ | $46.57_{\pm0.73}$ | $43.91_{\pm0.58}$ | $38.70_{\pm0.34}$ | $9.48_{\pm14.40}$ |
| PiSSA | 64 | $38.13_{\pm0.84}$ | $44.51_{\pm0.26}$ | $48.11_{\pm0.14}$ | $48.77_{\pm0.12}$ | $\mathbf{49.43}_{\pm0.19}$ | $49.09_{\pm0.16}$ | $48.44_{\pm0.12}$ | $47.10_{\pm0.32}$ | $43.84_{\pm0.28}$ | $39.66_{\pm0.36}$ | $34.37_{\pm0.39}$ | $27.18_{\pm1.29}$ | $17.35_{\pm0.57}$ |

*Table 6.* Performance of **Qwen3-0.6B** fine-tuned on **mathematical reasoning** tasks with **rank = 128**.

### E.2. Gemma-3-1B

#### E.2.1. MATHEMATICAL REASONING

| Methods | Batch Size | Learning Rate | | | | | | | | | | | |
|---|---|---|---|---|---|---|---|---|---|---|---|---|---|
| | | 1.1e-5 | 2e-5 | 3.6e-5 | 6.3e-5 | 1.1e-4 | 2e-4 | 3.6e-4 | 6.3e-4 | 1.1e-3 | 2e-3 | 3.6e-3 | 6.3e-3 |
| LoRA | 16 | $9.78_{\pm0.36}$ | $11.16_{\pm0.28}$ | $13.58_{\pm0.18}$ | $15.48_{\pm0.15}$ | $18.43_{\pm0.14}$ | $\mathbf{20.00}_{\pm0.26}$ | $19.93_{\pm0.65}$ | $17.99_{\pm0.55}$ | $11.71_{\pm0.49}$ | $1.52_{\pm0.19}$ | $1.27_{\pm0.59}$ | $1.07_{\pm0.27}$ |
| | 64 | $6.88_{\pm0.04}$ | $9.12_{\pm0.39}$ | $10.79_{\pm0.37}$ | $13.23_{\pm0.25}$ | $15.65_{\pm0.57}$ | $17.54_{\pm0.29}$ | $19.73_{\pm0.16}$ | $\mathbf{20.46}_{\pm0.79}$ | $19.83_{\pm0.91}$ | $13.33_{\pm0.81}$ | $1.48_{\pm0.48}$ | $0.00_{\pm0.00}$ |
| | 128 | $5.70_{\pm0.34}$ | $6.95_{\pm0.23}$ | $9.41_{\pm0.44}$ | $11.43_{\pm0.40}$ | $13.68_{\pm0.77}$ | $15.92_{\pm0.45}$ | $18.58_{\pm0.44}$ | $19.60_{\pm0.09}$ | $\mathbf{20.32}_{\pm0.28}$ | $16.95_{\pm2.70}$ | $0.09_{\pm0.16}$ | $0.00_{\pm0.00}$ |
| DoRA | 16 | $9.89_{\pm0.24}$ | $11.16_{\pm0.51}$ | $13.84_{\pm0.41}$ | $15.61_{\pm0.11}$ | $18.21_{\pm0.45}$ | $20.11_{\pm0.26}$ | $\mathbf{20.96}_{\pm0.57}$ | $18.34_{\pm0.20}$ | $11.90_{\pm0.29}$ | $4.89_{\pm0.99}$ | $0.93_{\pm0.12}$ | $1.16_{\pm0.15}$ |
| | 64 | $6.72_{\pm0.09}$ | $9.19_{\pm0.19}$ | $10.53_{\pm0.20}$ | $13.45_{\pm0.31}$ | $15.72_{\pm0.32}$ | $17.66_{\pm0.20}$ | $19.96_{\pm0.05}$ | $\mathbf{20.82}_{\pm0.32}$ | $19.87_{\pm0.91}$ | $13.53_{\pm1.64}$ | $1.52_{\pm0.45}$ | $0.34_{\pm0.23}$ |
| | 128 | $5.55_{\pm0.11}$ | $7.21_{\pm0.18}$ | $9.72_{\pm0.17}$ | $11.58_{\pm0.25}$ | $13.98_{\pm0.33}$ | $16.19_{\pm0.46}$ | $18.25_{\pm0.23}$ | $19.67_{\pm0.71}$ | $\mathbf{20.33}_{\pm0.64}$ | $12.86_{\pm10.03}$ | $0.13_{\pm0.23}$ | $0.02_{\pm0.03}$ |
| Init[AB] | 16 | $9.73_{\pm0.35}$ | $12.10_{\pm0.14}$ | $14.41_{\pm0.49}$ | $16.73_{\pm0.37}$ | $18.38_{\pm0.53}$ | $20.39_{\pm0.38}$ | $\mathbf{20.55}_{\pm0.40}$ | $18.34_{\pm0.48}$ | $11.94_{\pm0.31}$ | $1.48_{\pm0.24}$ | $1.16_{\pm0.31}$ | $1.45_{\pm0.17}$ |
| | 64 | $6.51_{\pm0.22}$ | $9.15_{\pm0.12}$ | $11.28_{\pm0.20}$ | $13.20_{\pm0.24}$ | $15.88_{\pm0.39}$ | $17.89_{\pm0.30}$ | $20.08_{\pm0.26}$ | $\mathbf{20.98}_{\pm0.33}$ | $19.31_{\pm0.75}$ | $13.97_{\pm0.03}$ | $2.74_{\pm3.83}$ | $0.07_{\pm0.12}$ |
| | 128 | $6.06_{\pm0.35}$ | $7.05_{\pm0.33}$ | $9.53_{\pm0.22}$ | $11.81_{\pm0.08}$ | $13.98_{\pm0.79}$ | $16.46_{\pm0.39}$ | $18.36_{\pm0.21}$ | $20.37_{\pm0.39}$ | $\mathbf{20.66}_{\pm0.39}$ | $17.85_{\pm0.84}$ | $4.40_{\pm7.46}$ | $0.00_{\pm0.00}$ |
| MiLoRA | 16 | $12.44_{\pm0.07}$ | $13.77_{\pm0.25}$ | $16.28_{\pm0.24}$ | $18.45_{\pm0.47}$ | $20.04_{\pm0.19}$ | $\mathbf{20.63}_{\pm0.67}$ | $19.40_{\pm0.80}$ | $15.72_{\pm0.49}$ | $10.22_{\pm0.42}$ | $2.03_{\pm0.95}$ | $1.35_{\pm0.43}$ | $1.56_{\pm0.65}$ |
| | 64 | $8.82_{\pm0.40}$ | $11.25_{\pm0.20}$ | $13.16_{\pm0.11}$ | $15.54_{\pm0.29}$ | $17.43_{\pm0.24}$ | $19.56_{\pm0.33}$ | $\mathbf{20.03}_{\pm0.59}$ | $19.60_{\pm0.78}$ | $17.93_{\pm0.90}$ | $13.65_{\pm0.07}$ | $4.97_{\pm0.40}$ | $0.00_{\pm0.00}$ |
| | 128 | $7.32_{\pm0.33}$ | $9.57_{\pm0.24}$ | $11.76_{\pm0.33}$ | $13.54_{\pm0.12}$ | $16.02_{\pm0.16}$ | $18.39_{\pm0.26}$ | $19.70_{\pm0.34}$ | $\mathbf{19.99}_{\pm0.66}$ | $19.53_{\pm0.47}$ | $16.83_{\pm0.73}$ | $7.45_{\pm1.00}$ | $0.57_{\pm0.81}$ |
| PiSSA | 16 | $14.30_{\pm0.18}$ | $16.10_{\pm0.27}$ | $18.31_{\pm0.12}$ | $19.90_{\pm0.21}$ | $\mathbf{20.61}_{\pm0.28}$ | $19.09_{\pm0.20}$ | $16.10_{\pm0.64}$ | $13.25_{\pm0.55}$ | $8.41_{\pm0.13}$ | $4.67_{\pm0.29}$ | $2.50_{\pm1.27}$ | $0.96_{\pm0.15}$ |
| | 64 | $11.11_{\pm0.05}$ | $13.67_{\pm0.17}$ | $15.56_{\pm0.33}$ | $18.11_{\pm0.23}$ | $19.52_{\pm0.48}$ | $\mathbf{20.68}_{\pm0.77}$ | $20.59_{\pm0.32}$ | $19.11_{\pm0.86}$ | $15.53_{\pm0.37}$ | $9.57_{\pm0.72}$ | $5.78_{\pm0.37}$ | $0.33_{\pm0.46}$ |
| | 128 | $9.42_{\pm0.38}$ | $11.80_{\pm0.28}$ | $14.40_{\pm0.11}$ | $16.23_{\pm0.38}$ | $18.60_{\pm0.21}$ | $19.61_{\pm0.44}$ | $\mathbf{20.65}_{\pm0.44}$ | $19.21_{\pm1.15}$ | $16.91_{\pm0.19}$ | $13.87_{\pm0.97}$ | $6.28_{\pm0.49}$ | $1.19_{\pm0.36}$ |

*Table 7.* Performance of **Gemma-3-1B** fine-tuned on **mathematical reasoning** tasks with **rank=128**.

### E.3. Llama-2-7B

#### E.3.1. MATHEMATICAL REASONING

| Methods | Batch Size | Learning Rate | | | | | | | | | |
|---|---|---|---|---|---|---|---|---|---|---|---|
| | | 2.00e-05 | 3.56e-05 | 6.32e-05 | 1.12e-04 | 2.00e-04 | 3.56e-04 | 6.32e-04 | 1.12e-03 | 2.00e-03 | 3.56e-03 |
| LoRA | 16 | 23.16 | 24.55 | 26.05 | 28.73 | 30.70 | 32.18 | **32.94** | 32.02 | 27.71 | 0.00 |
| | 128 | 16.12 | 18.60 | 21.46 | 23.49 | 25.91 | 28.21 | 30.31 | 32.28 | **32.78** | 1.97 |
| DoRA | 16 | 22.74 | 24.34 | 26.20 | 28.44 | 30.62 | **33.20** | 32.71 | 32.43 | 1.59 | 1.98 |
| | 128 | 16.07 | 18.98 | 21.57 | 23.70 | 26.16 | 28.54 | 30.20 | 32.80 | **33.62** | 0.00 |
| Init[AB] | 16 | 20.89 | 23.36 | 27.08 | 29.25 | 31.24 | **33.30** | 32.78 | 31.34 | 27.38 | 0.04 |
| | 128 | 15.79 | 17.64 | 20.17 | 22.96 | 25.42 | 28.03 | 30.45 | 32.32 | **32.96** | 31.08 |
| MiLoRA | 16 | 21.12 | 23.45 | 25.61 | 28.38 | 30.59 | 32.49 | **33.22** | 32.46 | 27.56 | 0.00 |
| | 128 | 15.72 | 18.51 | 20.70 | 22.58 | 25.32 | 26.76 | 29.87 | 31.48 | **33.55** | 0.26 |
| PiSSA | 16 | 22.66 | 26.30 | 28.20 | 30.12 | **31.91** | 31.62 | 30.57 | 28.76 | 0.84 | 0.92 |
| | 128 | 18.94 | 21.60 | 22.80 | 26.23 | 28.14 | 30.61 | 31.64 | **31.86** | 31.53 | 0.48 |

*Table 8.* Performance of **Llama-2-7B** fine-tuned on **mathematical reasoning** with **rank=8**.

| Methods | Batch Size | Learning Rate | | | | | | | | |
|---------|------------|---------|---------|---------|---------|---------|---------|---------|---------|---------|
| | | **2.00e-05** | **3.56e-05** | **6.32e-05** | **1.12e-04** | **2.00e-04** | **3.56e-04** | **6.32e-04** | **1.12e-03** | **2.00e-03** |
| LoRA | 16 | 25.03 | 27.77 | 29.67 | 32.11 | 33.73 | 33.84 | **34.18** | 27.01 | 1.31 |
| | 128 | 19.93 | 22.25 | 24.08 | 26.29 | 29.13 | 32.19 | 33.24 | **34.62** | 0.00 |
| DoRA | 16 | 25.41 | 27.35 | 29.78 | 31.96 | 34.14 | **35.16** | 33.26 | 28.92 | 0.55 |
| | 128 | 20.60 | 22.41 | 24.02 | 26.66 | 29.71 | 31.96 | 33.41 | **34.25** | 0.00 |
| Init[AB] | 16 | 24.06 | 27.68 | 28.71 | 32.52 | 34.10 | **34.92** | 34.12 | 27.16 | 0.77 |
| | 128 | 17.83 | 20.96 | 23.15 | 26.62 | 28.34 | 30.79 | 33.38 | **34.59** | 0.97 |
| MiLoRA | 16 | 24.53 | 27.23 | 29.23 | 31.44 | 33.97 | **34.85** | 33.85 | 28.35 | 0.62 |
| | 128 | 18.23 | 21.45 | 23.65 | 26.54 | 27.97 | 30.29 | 32.39 | **34.59** | 0.00 |
| PiSSA | 16 | 29.42 | 30.68 | 32.75 | **33.66** | 33.62 | 32.66 | 29.30 | 18.36 | 0.15 |
| | 128 | 23.98 | 27.02 | 29.44 | 30.01 | 32.89 | 33.42 | **34.31** | 32.92 | 0.65 |

*Table 9.* Performance of **Llama-2-7B** fine-tuned on **mathematical reasoning** with **rank=32**.

| Methods | Batch Size | Learning Rate | | | | | | | | |
|---------|------------|---------|---------|---------|---------|---------|---------|---------|---------|---------|
| | | **2.00e-05** | **3.56e-05** | **6.32e-05** | **1.12e-04** | **2.00e-04** | **3.56e-04** | **6.32e-04** | **1.12e-03** | **2.00e-03** |
| LoRA | 16 | 29.21 | 31.30 | 33.25 | 35.45 | **35.91** | 35.10 | 27.41 | 0.97 | 0.00 |
| | 128 | 22.69 | 24.95 | 27.79 | 30.74 | 32.62 | 34.85 | **35.66** | 0.00 | 0.00 |
| DoRA | 16 | 29.43 | 30.75 | 33.14 | 35.73 | **36.41** | 34.54 | 1.28 | 0.94 | 0.79 |
| | 128 | 23.27 | 25.63 | 27.87 | 30.11 | 33.00 | 35.10 | **35.57** | 0.38 | 0.00 |
| Init[AB] | 16 | 29.04 | 31.52 | 31.96 | 34.81 | **36.72** | 35.41 | 27.98 | 1.54 | 0.00 |
| | 128 | 22.01 | 25.03 | 28.11 | 30.47 | 31.80 | 34.78 | **35.57** | 34.45 | 0.00 |
| MiLoRA | 16 | 28.23 | 31.11 | 33.42 | 35.22 | **36.02** | 34.71 | 28.03 | 0.38 | 0.00 |
| | 128 | 22.14 | 24.84 | 27.88 | 30.33 | 31.29 | 33.67 | **35.23** | 0.00 | 0.00 |
| PiSSA | 16 | 33.35 | 35.03 | 35.27 | **35.83** | 32.89 | 27.90 | 16.75 | 1.45 | 0.00 |
| | 128 | 29.84 | 31.64 | 33.44 | 34.45 | **35.31** | 34.99 | 31.59 | 27.83 | 0.00 |

*Table 10.* Performance of **Llama-2-7B** fine-tuned on **mathematical reasoning** with **rank=128**.

### E.3.2. CODE GENERATION

| Methods | Batch Size | Learning Rate | | | | | | | | | |
|---|---|---|---|---|---|---|---|---|---|---|---|
| | | 2.00e-05 | 3.56e-05 | 6.32e-05 | 1.12e-04 | 2.00e-04 | 3.56e-04 | 6.32e-04 | 1.12e-03 | 2.00e-03 | 3.56e-03 |
| LoRA | 16 | 27.20 | 29.40 | 27.55 | 30.00 | 31.40 | 33.25 | **36.10** | 32.65 | 32.30 | 0.00 |
| | 128 | 24.50 | 25.60 | 27.05 | 28.15 | 29.30 | 30.90 | 30.45 | **35.35** | 33.35 | 0.00 |
| DoRA | 16 | 28.05 | 28.45 | 28.90 | 31.15 | 31.80 | 33.15 | **33.80** | 32.00 | 29.50 | 0.00 |
| | 128 | 24.65 | 26.95 | 27.25 | 29.65 | 29.15 | 30.15 | 33.15 | 33.15 | **33.35** | 32.55 |
| Init[AB] | 16 | 26.25 | 27.30 | 30.30 | 29.90 | 32.90 | 34.15 | **35.60** | 32.70 | 31.20 | 0.00 |
| | 128 | 23.00 | 25.50 | 25.55 | 29.15 | 30.50 | 32.05 | 33.55 | **34.20** | 33.25 | 0.00 |
| MiLoRA | 16 | 26.80 | 28.60 | 27.55 | 28.75 | 31.60 | **33.95** | 32.25 | 33.30 | 29.85 | 0.00 |
| | 128 | 22.50 | 25.50 | 26.40 | 27.20 | 29.55 | 29.30 | 32.35 | 32.40 | **33.35** | 0.00 |
| PiSSA | 16 | 29.05 | 29.90 | 32.55 | 31.90 | **34.05** | 30.75 | 29.85 | 29.60 | 24.35 | 0.00 |
| | 128 | 27.45 | 27.70 | 30.15 | 30.45 | 31.45 | 30.10 | **33.55** | 31.75 | 30.75 | 29.35 |

*Table 11.* Performance of **Llama-2-7B** fine-tuned on **code generation** with **rank=8**.

| Methods | Batch Size | Learning Rate | | | | | | | | |
|---|---|---|---|---|---|---|---|---|---|---|
| | | 2.00e-05 | 3.56e-05 | 6.32e-05 | 1.12e-04 | 2.00e-04 | 3.56e-04 | 6.32e-04 | 1.12e-03 | 2.00e-03 |
| LoRA | 16 | 28.30 | 29.10 | 30.55 | 35.30 | 37.05 | **37.65** | 36.85 | 28.75 | 0.00 |
| | 128 | 27.60 | 27.25 | 29.25 | 29.30 | 30.85 | 34.55 | 35.45 | **35.65** | 32.20 |
| DoRA | 16 | 29.00 | 29.10 | 30.30 | 34.75 | 34.20 | 36.40 | **37.40** | 29.35 | 0.00 |
| | 128 | 26.75 | 27.15 | 29.00 | 30.30 | 31.20 | 35.25 | **35.45** | 35.15 | 0.00 |
| Init[AB] | 16 | 27.75 | 29.70 | 31.10 | 34.05 | **35.40** | 35.20 | 33.05 | 31.25 | 0.00 |
| | 128 | 26.10 | 27.00 | 27.75 | 29.00 | 31.60 | 35.70 | **37.00** | 36.70 | 0.00 |
| MiLoRA | 16 | 27.15 | 28.55 | 30.55 | 32.75 | 34.10 | 35.90 | **36.10** | 27.40 | 0.00 |
| | 128 | 25.70 | 26.50 | 26.90 | 28.90 | 31.05 | 31.40 | 33.30 | **34.75** | 0.00 |
| PiSSA | 16 | 32.10 | 33.00 | 34.00 | 32.10 | **34.45** | 32.30 | 27.75 | 18.55 | 0.00 |
| | 128 | 29.80 | 30.05 | 31.75 | 32.10 | 33.55 | **35.00** | 33.20 | 32.60 | 0.00 |

*Table 12.* Performance of **Llama-2-7B** fine-tuned on **code generation** tasks with **rank = 32**.

| Methods | Batch Size | Learning Rate | | | | | | | | |
|---|---|---|---|---|---|---|---|---|---|---|
| | | 2.00e-05 | 3.56e-05 | 6.32e-05 | 1.12e-04 | 2.00e-04 | 3.56e-04 | 6.32e-04 | 1.12e-03 | 2.00e-03 |
| LoRA | 16 | 30.72 | 31.87 | 34.23 | **37.55** | 37.40 | 36.68 | 29.20 | 0.00 | 0.00 |
| | 128 | 29.37 | 29.82 | 31.18 | 33.40 | 35.70 | 36.48 | **36.68** | 13.05 | 0.00 |
| DoRA | 16 | 30.93 | 32.50 | 34.72 | 36.70 | **38.30** | 35.82 | 30.50 | 0.00 | 0.00 |
| | 128 | 29.12 | 30.17 | 31.22 | 32.80 | 36.12 | **38.12** | 37.50 | 0.00 | 0.00 |
| Init[AB] | 16 | 30.73 | 31.48 | 34.47 | 36.75 | **38.07** | 35.82 | 30.72 | 0.00 | 0.00 |
| | 128 | 28.32 | 30.00 | 30.15 | 32.80 | 35.83 | 36.97 | **38.43** | 0.00 | 0.00 |
| MiLoRA | 16 | 29.72 | 32.18 | 33.55 | 36.53 | **37.08** | 35.95 | 30.03 | 0.00 | 0.00 |
| | 128 | 28.67 | 29.20 | 29.72 | 31.93 | 34.67 | **37.42** | 37.07 | 0.00 | 0.00 |
| PiSSA | 16 | 35.47 | **37.35** | 35.98 | 36.08 | 33.92 | 28.43 | 17.53 | 0.00 | 0.00 |
| | 128 | 31.90 | 32.33 | 34.80 | 35.42 | **36.77** | 36.67 | 34.50 | 26.90 | 0.00 |

*Table 13.* Performance of **Llama-2-7B** fine-tuned on **code generation** tasks with **rank = 128**.

# F. Example Model Responses

We examine the responses of Gemma-3-1B fine-tuned on the mathematical reasoning task using LoRA and PiSSA ($r = 128, B = 16$) under various learning rates. Figure 9 presents a randomly selected testing example from the MATH dataset, with the corresponding model responses organized in Table 14. In this example, it is easily observed that the two PEFT methods operate in distinct effective learning rate regimes. Specifically, LoRA produces correct reasoning paths within the range of $2.0 \times 10^{-4}$ to $6.3 \times 10^{-4}$, whereas PiSSA achieves accurate results in a slightly lower range of $6.3 \times 10^{-5}$ to $2.0 \times 10^{-4}$. Moreover, the LLM under PiSSA fine-tuning tends to continue responding reasonably at larger learning rates, as evident when $\eta = 2.0 \times 10^{-3}$ and $3.6 \times 10^{-3}$, where the model produces answers of "6" and "4", while under LoRA fine-tuning, the model diverges to output repetitive gibberish. These qualitative findings correspond to our discussion in Sec. 4.3.1.

---

**Selected MATH Testing Set Input**

Below is an instruction that describes a task. Write a response that appropriately completes the request.

### Instruction:
There are eight boys and six girls who are members of the trumpet section in the band. From the members of the trumpet section, a quintet is being formed. If the quintet must consist of three boys and two girls, how many quintets are possible?

### Response: Let's think step by step.

---

*Figure 9.* An example input from the MATH dataset used for testing.

*Table 14.* Qualitative comparison of LoRA and PiSSA reasoning paths across learning rates.

| LR ($\eta$) | LoRA Response | PiSSA Response |
|---|---|---|
| $1.12 \times 10^{-5}$ | We are given that there are 8 boys and 6 girls in the trumpet section. To form a quintet, we need to choose 3 boys and 2 girls. So, we need to choose 3 boys from the 8 boys and 2 girls from the 6 girls. This can be done in $\binom{8}{2} = \frac{8!}{2!6!} = \frac{8 \cdot 7}{2 \cdot 1} = \boxed{16}$ ways. The answer is: 16 
 ✗ **Incorrect** | We have 8 boys and 6 girls, so we need to choose 3 boys from the 8 boys and 2 girls from the 6 girls. This can be done in $\binom{8}{3} \cdot \binom{6}{2} = \frac{8!}{3!3!} \cdot \frac{6!}{2!2!} = \frac{8 \cdot 7}{3 \cdot 2} = \boxed{14}$ ways. The answer is: 14 
 ✗ **Incorrect** |
| $2.00 \times 10^{-5}$ | We are given that there are 8 boys and 6 girls in the trumpet section. To form a quintet, we need to choose 3 boys and 2 girls. So, the number of quintets that can be formed is $\frac{8}{3} \times \frac{6}{2} = \frac{8 \times 6}{3 \times 2} = \frac{48}{6} = \boxed{8}$. The answer is: 8 
 ✗ **Incorrect** | We have 8 boys and 6 girls, so we need to choose 3 boys from the 8 boys and 2 girls from the 6 girls. This can be done in $\binom{8}{3} \cdot \binom{6}{2} = \frac{8!}{3!3!} = \frac{8 \cdot 7}{3 \cdot 2} = 28$ ways. Therefore, there are $\boxed{28}$ quintets possible. The answer is: 28 
 ✗ **Incorrect** |
| $3.56 \times 10^{-5}$ | We have 8 boys and 6 girls, so there are a total of $8 + 6 = 14$ members in the trumpet section. To form a quintet, we need to choose 3 boys and 2 girls from the 14 members. The number of quintets that can be formed is the number of ways to choose 3 boys and 2 girls from a group of 14 members. Using the combination formula, this is $\binom{14}{3} = \frac{14!}{3!11!} = \frac{14 \times 13 \times 12}{3 \times 2 \times 1} = \boxed{105}$. The answer is: 105 
 ✗ **Incorrect** | We have 8 boys and 6 girls in the trumpet section. To form a quintet, we need to choose 3 boys and 2 girls from the 8 boys and 6 girls. This can be done in $\binom{8}{3} \cdot \binom{6}{2} = \frac{8!}{3!3!} = \frac{8 \cdot 7}{3 \cdot 2} = 28$ ways. Therefore, there are $\boxed{28}$ possible quintets. The answer is: 28 
 ✗ **Incorrect** |

| LR ($\eta$) | LoRA Response | PiSSA Response |
|---|---|---|
| $6.32 \times 10^{-5}$ | We have 8 boys and 6 girls, so we need to choose 3 boys from the 8 boys and 2 girls from the 6 girls.
This can be done in
$\binom{8}{3} \cdot \binom{6}{2} = \frac{8!}{3!3!} \cdot \frac{6!}{2!4!} = \frac{8 \cdot 7 \cdot 6}{3 \cdot 2 \cdot 1} = \frac{216}{6} = \boxed{36}$.
The answer is: 36

✗ **Incorrect** | We need to choose 3 boys from the 8 members of the trumpet section, and 2 girls from the 6 members of the trumpet section.
The number of ways to choose 3 boys from 8 is $\binom{8}{3} = 56$.
The number of ways to choose 2 girls from 6 is $\binom{6}{2} = 15$.
Therefore, the total number of quintets is $56 \times 15 = \boxed{840}$.
The answer is: 840
✓ **Correct** |
| $1.12 \times 10^{-4}$ | We have 8 boys and 6 girls, so we need to choose 3 boys from the 8 boys and 2 girls from the 6 girls.
This can be done in
$\binom{8}{3} \cdot \binom{6}{2} = \frac{8!}{3!3!} \cdot \frac{6!}{2!2!} = \frac{8 \cdot 7 \cdot 6}{3 \cdot 2 \cdot 1} = 56$ ways.
Therefore, there are $\boxed{56}$ possible quintets.
The answer is: 56

✗ **Incorrect** | We need to choose 3 boys from the 8 available boys, and 2 girls from the 6 available girls.
The number of ways to choose 3 boys from 8 is $\binom{8}{3} = 56$.
The number of ways to choose 2 girls from 6 is $\binom{6}{2} = 15$.
Therefore, the total number of quintets is $56 \times 15 = \boxed{840}$.
The answer is: 840
✓ **Correct** |
| $2.00 \times 10^{-4}$ | We need to choose 3 boys from the 8 available, and 2 girls from the 6 available.
This can be done in $\binom{8}{3} = 56$ ways.
Similarly, we need to choose 2 girls from the 6 available, and 3 boys from the 3 available.
This can be done in $\binom{6}{2} = 15$ ways.
Therefore, there are $56 \times 15 = \boxed{840}$ possible quintets.
The answer is: 840
✓ **Correct** | First, we choose 3 boys from the 8 available boys. This can be done in $\binom{8}{3} = 56$ ways.
Next, we choose 2 girls from the 6 available girls. This can be done in $\binom{6}{2} = 15$ ways.
Therefore, the total number of quintets is $56 \times 15 = \boxed{840}$.
The answer is: 840

✓ **Correct** |
| $3.56 \times 10^{-4}$ | We need to choose 3 boys from the 8 available, and 2 girls from the 6 available.
The number of ways to choose 3 boys from 8 is $\binom{8}{3} = 56$.
The number of ways to choose 2 girls from 6 is $\binom{6}{2} = 15$.
Therefore, the total number of quintets is $56 \times 15 = \boxed{840}$.
The answer is: 840
✓ **Correct** | For the boys, we have 8 choices for the quintet.
For the girls, we have 6 choices for the quintet.
So, the total number of quintets is $8 \times 6 = \boxed{48}$.
The answer is: 48

✗ **Incorrect** |
| $6.32 \times 10^{-4}$ | We have 8 boys and 6 girls, so we need to choose 3 boys and 2 girls.
The number of ways to choose 3 boys from 8 is $\binom{8}{3} = 56$.
The number of ways to choose 2 girls from 6 is $\binom{6}{2} = 15$.
Therefore, the total number of quintets is $56 \times 15 = \boxed{840}$.
The answer is: 840
✓ **Correct** | There are 8 boys and 6 girls, so there are a total of 8 + 6 = 14 members in the trumpet section.
We need to choose 3 boys from the 8 boys, so there are 8 boys.
We also need to choose 2 girls from the 6 girls, so there are 6 girls.
Therefore, the total number of quintets is $8 + 3 + 6 = \boxed{17}$.
The answer is: 17
✗ **Incorrect** |

| LR ($\eta$) | LoRA Response | PiSSA Response |
|---|---|---|
| $1.12 \times 10^{-3}$ | From the members of the trumpet section, there are 8 boys and 6 girls.
From the members of the quintet, we need to choose 3 boys from the 8 boys and 2 girls from the 6 girls.
This can be done in $\binom{8}{3} = \frac{8!}{3!5!} = \frac{8\cdot7\cdot6}{3\cdot2\cdot1} = \boxed{35}$ ways.
The answer is: 35
✗ **Incorrect** | There are 8 boys and 6 girls in the band, so there are a total of $8+6=14$ members in the band.
We want to find the number of quintets that are possible.
Since the quintet must consist of three boys and two girls, we need to choose 3 boys out of the 14 members.
This can be done in $\binom{14}{3} = 46$ ways.
Therefore, there are $46-14 = \boxed{32}$ possible quintets.
The answer is: 32
✗ **Incorrect** |
| $2.00 \times 10^{-3}$ | Since the number of boys is x, we can set up the equation:
Number of boys = x
Number of boys with the number of boys with the two boys with the two boys with the two boys with the two boys...(repetitive gibberish)
✗ **Incorrect** | There are 8 boys and 6 girls, so there are a total of 8 + 6 = 14 people in the band.
There are 6 girls and 2 girls, so there are a total of 6 + 2 = 8 people in the band.
There are 8 quintet members in the band.
Therefore, the number of quintets possible is 14 - 8 = 6.
#### 6
The answer is: 6
✗ **Incorrect** |
| $3.56 \times 10^{-3}$ | The total number of students is 1, so the number of people is 1, 1, 1, 1, 1, 1, 1, 1, 1, 1, 1, 1, 1, 1, 1, 1, 1, 1, 1, 1, 1, 1, 1, 1, 1, 1, 1, 1...(repetitive gibberish)
✗ **Incorrect** | There are 8 boys and 6 girls, so there are a total of $8+6=14$ possible outcomes. Out of the 14 girls, 6 are girls, so there are $14-6=18$ possible outcomes.
Therefore, there are $18-14 = \boxed{4}$ possible outcomes.
The answer is: 4
✗ **Incorrect** |
| $6.32 \times 10^{-3}$ | The number of the first circle is 12, 10 = 10.
The total number of the number of the number of the number of the number of the number of the number of the first two sides of the first two sides.
The area of the first two sides is 10 + 1 = 120 = 100
The total number of the first two sides is 1000.
The total number of the number of the first two sides is 1000, so the total number of the first 100000.
The answer is 10000000000...(repetitive gibberish)
✗ **Incorrect** | We know that the total amount of the water is 1.
We also know that the total number of eggs is 1.
The total number of boys who has not the number of students is 1.
We can set up the equation as follows: Number of students who has not the pool + Number of students who has not the number of students who in the pool...(repetitive gibberish)
✗ **Incorrect** |

# G. Hessian Computation Details

We select the Lanczos algorithm over the Power Iteration method for the presented eigenvalue problem, as the latter converges to eigenvalues in descending order of magnitude, whereas our focus is on probing the algebraically largest eigenvalue of the Hessian. Sec. G.1 explains implementation details of the Lanczos algorithm, and Sec. G.2 provides additional Hessian results.

## G.1. Lanczos Algorithm Implementation Details

Our implementation is based on several Hessian-related codebases, such as PyHessian[9] (Yao et al., 2020) and LLM-Hessian[10] (Ilin, 2025), with modifications to suit our custom scenario. Algorithm 1 summarizes our implementation of the Lanczos Algorithm for estimating $\lambda_{\max}(\mathbf{H})$. We set the Lanczos iterations $m = 100$ and tolerance $\epsilon = 5 \times 10^{-3}$. At each Lanczos iteration step, the Hessian-Vector Product (HVP) is applied to calculate $\mathbf{H}\mathbf{q}_k$ without explicitly forming $\mathbf{H}$ (Algorithm 2).

We strictly ensure that the loss is calculated identically to that in supervised fine-tuning, rendering the resulting curvature information meaningful. In particular, we ensure that (1) the input prompt (i.e., instruction or question) tokens are masked out from the loss calculation, and (2) the loss is averaged over each token instead of each sentence[11]. To ensure computational feasibility, a subset of $N = 500$ training samples from MetaMathQA is selected for loss calculation, and a batch size of $B = 5$ is utilized. Figure 10 validates that this sample size is sufficient for reliably estimating the Hessian of the downstream task. Due to the numerical instability nature of the Lanczos algorithm, both the base model and adapters are in Float32 precision.

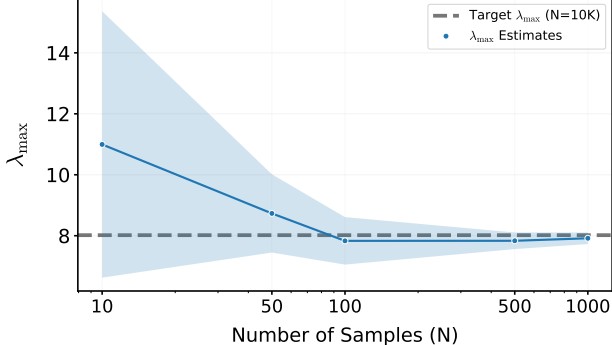

*Figure 10.* Approximately 500 training samples are sufficient for stable Hessian estimation. The figure reports the estimated $\lambda_{\max}$ for PiSSA on the first Query projection matrix of Qwen3-0.6B ($r = 128$). We track these estimates across varying sample sizes from the MetaMathQA dataset, using N=10k as the reference baseline. Results represent the mean and standard deviation over 5 randomly selected subsets for each $N$.

---

[9]https://github.com/amirgholami/PyHessian

[10]https://github.com/vectozavr/llm-hessian

[11]This is the default way of calculating loss during LLM supervised fine-tuning; see https://github.com/huggingface/transformers/issues/34510

---

**Algorithm 1** Estimating Maximum Eigenvalues of Hessian by Lanczos Iterations

---

**Input:** LoRA parameters $\theta = \{B_0, A_0\}$, downstream dataset $\mathcal{D}$, iterations $m$, initial vector $\mathbf{b}$, tolerance $\epsilon$.
**Output:** Approximation of the maximum eigenvalue $\lambda_{\max}(\mathbf{H})$.
**Initialization:**
Set $\beta_0 = 0$, $\mathbf{q}_0 = \mathbf{0}$, $\lambda_{\text{prev}} = -\infty$.
Normalize initial vector: $\mathbf{q}_1 = \mathbf{b}/\|\mathbf{b}\|_2$.
**Lanczos Iteration:**
**for** $k = 1$ **to** $m$ **do**
    Compute Hessian-Vector Product:
    $\mathbf{v} = \text{HVP}(\theta, \mathcal{D}, \mathbf{q}_k)$         // See Algorithm 2
    Compute diagonal element of $T$:
    $\alpha_k = \mathbf{q}_k^\top \mathbf{v}$
    Orthogonalize (Gram-Schmidt):
    $\mathbf{v} = \mathbf{v} - \beta_{k-1}\mathbf{q}_{k-1} - \alpha_k\mathbf{q}_k$
    Compute off-diagonal element of $T$:
    $\beta_k = \|\mathbf{v}\|_2$
    **Convergence Check:**
    Construct symmetric tridiagonal matrix $T_k \in \mathbb{R}^{k \times k}$ using $\alpha_{1:k}, \beta_{1:k-1}$.
    `eig_vals` $\leftarrow$ `torch.linalg.eigvalsh`$(T_k)$
    $\lambda_{\text{curr}} = \max(\text{eig\_vals})$
    **if** $|\lambda_{\text{curr}} - \lambda_{\text{prev}}| < \epsilon$ **then**
        Return $\lambda_{\text{curr}}$
    **end if**
    $\lambda_{\text{prev}} \leftarrow \lambda_{\text{curr}}$
    **if** $\beta_k \approx 0$ **then**
        Return $\lambda_{\text{curr}}$         // The Krylov subspace is invariant
    **end if**
    Normalize: $\mathbf{q}_{k+1} = \mathbf{v}/\beta_k$
**end for**
**Return** $\lambda_{\text{curr}}$ =0

---

**Algorithm 2** Hessian-Vector Product (HVP) Calculation

---

**Input:** LoRA parameters $\theta = \{B_0, A_0\}$, downstream dataset $\mathcal{D}$, vector $\mathbf{v}$, sample size $N$, batch size $B$.
**Output:** The Hessian-Vector product $\mathbf{Hv}$.
**Sampling:**
Sample a subset $\mathcal{S}$ of size $N$ from $\mathcal{D}$.
**Initialization:**
Set accumulator $\mathbf{u} = \mathbf{0}$.
Set total token counter $C_{\text{total}} = 0$.
**Batch Processing:**
**for** each mini-batch $\mathcal{B}$ of size $B$ from $\mathcal{S}$ **do**
    Count supervised tokens in batch: $c_{\mathcal{B}} = \text{CountTokens}(\mathcal{B})$.
    $C_{\text{total}} \leftarrow C_{\text{total}} + c_{\mathcal{B}}$.
    **Forward Pass for Loss Calculation:**
    Compute sum of Cross-Entropy losses over all supervised tokens in $\mathcal{B}$:
    $\mathcal{L}_{\text{batch}}(\theta) = \sum_{(x,y)\in\mathcal{B}} \ell(f(x;\theta), y)$         // `torch.nn.CrossEntropyLoss(sum)`
    **Double Backward for HVP:**
    $\mathbf{g} = \nabla_\theta \mathcal{L}_{\text{batch}}$
    $s = \mathbf{g}^\top \mathbf{v}$
    $\mathbf{h}_{\mathcal{B}} = \nabla_\theta s$         // Implemented via `torch.autograd.functional.vhp`
    **Accumulate:**
    $\mathbf{u} \leftarrow \mathbf{u} + \mathbf{h}_{\mathcal{B}}$         // Summing up batch-wise contributions
**end for**
**Normalize:**
$\mathbf{u} \leftarrow \mathbf{u}/C_{\text{total}}$         // Average over total supervised tokens
**Return** $\mathbf{u}$ =0

---

### G.2. Additional Hessian Results

Figure 11 presents the detailed $\lambda_{\max}$ values of the Query projection matrix for Qwen across Transformer layers, providing a layer-wise breakdown of the results shown in Figure 6. We observe intriguing patterns where all methods tend to have high or low values at similar layer locations. For example, at layer 20, $\lambda_{\max} = \{4.7, 8.5, 8.3, 53.8\}$ for LoRA, Init[AB], MiLoRA, and PiSSA, respectively; whereas at layer 26, the values drop to $\{0.2, 0.7, 0.7, 2.3\}$. However, at any given layer, PiSSA consistently exhibits a larger $\lambda_{\max}$ compared to LoRA, by around an order of magnitude.

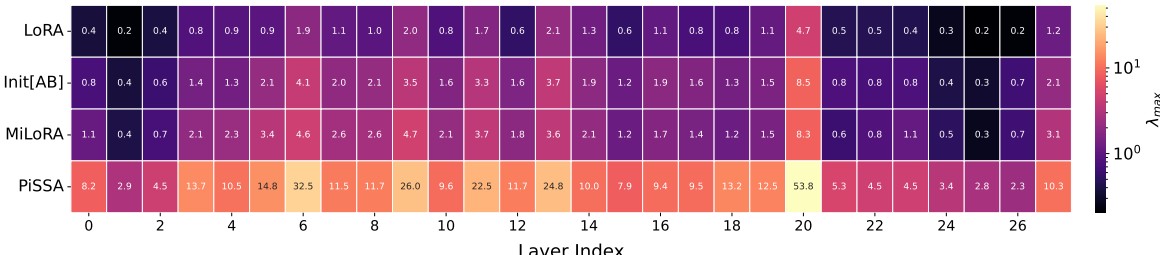

*Figure 11.* Heatmap of the top eigenvalues of the Query projection matrix across Transformer layers, i.e., $\lambda_{\max}^{Q,i}$ for $i = 1, \ldots, L$, for Qwen3-0.6B on a mathematical reasoning task ($r = 128$). All methods exhibit similar distributional patterns, with PiSSA consistently exhibiting significantly larger values compared to vanilla LoRA.

## H. Limitations and Future Work

In this paper, we focused our investigation on decoder-only LLMs up to the 7B parameter scale and restricted our evaluation to mathematical and code generation tasks. Hence, the scalability of our findings to larger foundation models and their generalization to diverse downstream tasks remain to be verified. Additionally, computational costs required for hyperparameter searches precluded the exhaustive tuning of all training configurations, such as learning rate schedulers, warmup steps, and LoRA adapter placements. It is possible that fine-grained tuning of these configurations could yield further performance gains or distinct convergence behaviors.

We also highlight that our findings may not extend to untested model architectures (e.g., encoder-only LLMs (Devlin et al., 2019), Vision Transformers (Dosovitskiy, 2020), and Vision-Language Models (Alayrac et al., 2022)) or to all existing advanced LoRA variants. For instance, several methods have originally reported higher peak performance than LoRA under comprehensive learning rate sweeps—such as LoRA-One (Zhang et al., 2025g), which initializes adapters via the SVD of the one-step full gradient, with $\approx 2\%$ performance improvement on Llama (cf. Zhang et al. (2025g) Table 3). Moreover, fine-tuned accuracy on standard benchmarks is not the sole criterion for evaluating PEFT algorithms; specific variants may offer distinct advantages in other dimensions, e.g., mitigating catastrophic forgetting of pretrained knowledge (Biderman et al., 2024; Wang et al., 2024b; Yang et al., 2024a; Zhang et al., 2025c; Xiong & Xie, 2025).

While the landscape of LoRA variants continues to expand, our results suggest that vanilla LoRA already suffices as a competitive baseline, potentially indicating that weight-based low-rank adaptation strategies may be approaching saturation. Looking ahead, we posit that further investigation into alternative adaptation mechanisms may unlock new dimensions of efficiency. Examples of such mechanisms include hidden representation fine-tuning (Wu et al., 2024; Yin et al., 2024) and approaches that adapt non-linear functions within layers (Yin et al., 2025). We leave the exploration of these orthogonal paradigms as future work.

