# OpenReview forum: "Learning Rate Matters: Vanilla LoRA May Suffice for LLM Fine-tuning"
_ICML.cc/2026/Conference — Submitted to ICML 2026_

### Official Review · Reviewer_txFF · 2026-03-06

**Soundness:** 3
**Presentation:** 3
**Significance:** 3
**Originality:** 3
**Overall Recommendation:** 5
**Confidence:** 4

**Summary:**

This paper systematically revisits comparisons between several LoRA variants and vanilla LoRA, highlighting that the learning rate is a key confounding factor: different methods prefer substantially different learning-rate ranges, yet after proper tuning their peak performance is often very close. The paper further uses the largest Hessian eigenvalue to explain the observed negative correlation between optimal learning rate and curvature, as well as PiSSA’s need for a lower learning rate.

**Compliance With Llm Reviewing Policy:**

Affirmed.

**Final Justification:**

The task addressed in this paper is important, and I am glad to see work emerging in this topic. My main concern with the paper is that the method is not yet sufficiently systematic, and there is still room to further strengthen the experimental evaluation. That said, within the limited rebuttal period, the authors addressed these concerns reasonably well, and their additional responses have increased my confidence that they are capable of providing more comprehensive experiments. Accordingly, I have raised my score.

**Key Questions For Authors:**

See Weaknesses.

**Limitations:**

yes

**Strengths And Weaknesses:**

**Strengths**
1. The topic is important. The paper points out a common issue in parameter-efficient fine-tuning—claims derived from fixed or narrowly tuned hyperparameters—which makes conclusions potentially non-robust. This analysis is a significant contribution to the community.
2. The writing is clear, the figures and tables are well presented, and the appendix provides sufficient additional details and evidence.
3. The conclusions are insightful and supported by fairly extensive experiments.

**Weaknesses**
1. I understand the experimental cost is high, and the current coverage is relatively broad. However, the evaluation still lacks some broader settings, such as general QA tasks, reasoning models, larger base models, and more LoRA variants.
2. The depth of analysis could be improved. The paper mainly enumerates several LoRA algorithms without organizing them into a more structured taxonomy, e.g., characterizing what distinctive properties different “optimization angles” bring. Strengthening this part would further elevate the contribution.

---

> ### Author Rebuttal · Authors · 2026-03-31
>
> We are pleased to know that the reviewer considers our analysis a significant contribution to the community, and we thank the reviewer for the positive feedback on our presentation and conclusions. We address the reviewer's concerns as follows:
>
> **Weakness 1**
> We appreciate the reviewer's understanding regarding the computational cost of the experiments. To better support our conclusions, we have followed the reviewer's advice to scale up our evaluation to models with up to 13B parameters (Llama-2-13B) and include general commonsense reasoning tasks, as well as five additional recently published LoRA variants: EVA, LoRA-GA, GraLoRA, LoRA-Pro, RandLoRA. This results in a total coverage of **four models, three task types, and ten LoRA methods (1x vanilla and 9x variants)**, as shown below in Table R5.
>
> Specifically, the commonsense reasoning tasks consist of eight subtasks (i.e., BoolQ, PIQA, SIQA, HellaSwag, WinoGrande, ARC-e, ARC-c, and OBQA), which are all structured in general QA style. Table R5 shows the average performance across these eight subtasks on Gemma-3-1B when fine-tuning with the ten methods. Notably, all methods, including the originally considered and the newly added ones, peak at a similar performance level ($\approx$ 37%), while their optimal learning rate ranges differ. For example, PiSSA, LoRA-GA, EVA, and RandLoRA perform well at lower learning rates (e.g., reaching >30% at 2.00e-06), while GraLoRA and LoRA-Pro tend to operate at higher ones (e.g., achieving >20% at 2.00e-02). We refer the reviewer to our response [posted under reviewer pJ4H](https://openreview.net/forum?id=1M64XF14oc&noteId=lJbvKNm2SV), where we provide further details on the experimental setup and rationales for selecting these LoRA variants.
>
> **Table R5** Performance of Gemma-3-1B on commonsense reasoning tasks ($r=4$, $B=64$). The results are averaged across eight subtasks. Both the originally considered methods and the newly added variants are presented. All values are reported with means and stds over three training runs, with the best performance highlighted in **bold**.
>
> |Methods|2.00e-06|6.32e-06|2.00e-05|6.32e-05|2.00e-04|6.32e-04|2.00e-03|6.32e-03|2.00e-02|
> |-|-|-|-|-|-|-|-|-|-|
> |**LoRA**| 10.07±0.08 | 15.38±0.08 | 36.85±0.24 | 37.05±0.19 | **37.40±0.24** | 37.40±0.06 | 37.04±0.25 | 17.03±19.04 | 0.00±0.00|
> |**DoRA**|10.51±0.14|20.66±0.14|37.06±0.10|37.33±0.15|37.32±0.16|**37.43±0.09**|37.08±0.03|37.03±0.78|1.45±2.52|
> |**Init[AB]**|11.90±0.12|22.70±0.53|37.07±0.28 |37.28±0.15|**37.26±0.26**|37.13±0.11|36.95±0.14|37.18±0.35|4.49±7.77|
> |**MiLoRA**|14.33±0.14|36.65±0.01|**37.59±0.07**|37.53±0.21|37.23±0.11|37.06±0.23|36.85±0.18|37.13±0.04|5.32±7.22|
> |**PiSSA**|36.16±0.05|37.02±0.35|36.99±0.19|36.94±0.24|**37.12±0.29**|37.03±0.22|36.67±0.16|36.86±1.21|4.55±7.87|
> |**EVA**|33.46±0.04|36.98±0.06|37.08±0.17|37.18±0.09| 37.19±0.24|**37.20±0.20**|36.88±0.15|36.95±0.55|0.00±0.00|
> |**LoRA-GA**|36.99±0.06|37.16±0.32|37.05±0.21|37.01±0.23|**36.93±0.11**| 36.78±0.09|36.81±0.20|36.81±0.09|10.97±19.01|
> |**GraLoRA**|8.32±1.50|11.70±0.10|36.97±0.05|37.38±0.08|**37.59±0.31**|37.15±0.39|37.10±0.03|36.93±0.09|22.89±11.29|
> |**LoRA-Pro**|15.98±0.09|36.93±0.05|36.94±0.06|36.95±0.09|**37.19±0.23**|36.96±0.13|36.89±0.13|36.86±0.36|26.13±11.71|
> |**RandLoRA**|37.13±0.11|37.10±0.18|37.06±0.31|**37.20±0.18**|36.87±0.32|36.88±0.50|37.02±0.27|36.66±1.17|36.92±0.09|
>
> For the Llama-2-13B experiments, we refer the reviewer to Tables R1 and R2 ([posted under reviewer kqpA](https://openreview.net/forum?id=1M64XF14oc&noteId=yyCfpVhJo4)), which present the mathematical reasoning results for rank 8 and rank 128, respectively. While we agree with the reviewer that evaluating fine-tuning on reasoning models could be a potential addition, we argue that using such models is not a common setup in PEFT research, as evident in Table 2 of our paper, where none of the 52 surveyed methods fine-tune reasoning models. We therefore defer the evaluation of reasoning models to future work.
>
> **Weakness 2**
> We thank the reviewer for raising this point. We are committed to leveraging the additional page available in the final revision to provide a more organized taxonomy of LoRA variants before introducing their algorithms, while highlighting their distinctive optimization mechanisms and properties.

---

> > ### Author Rebuttal · Reviewer_txFF · 2026-04-01
> >
> > The authors provide a relatively comprehensive experimental response to the first weakness, but the second weakness is not fully addressed. However, given the limited time and space available during the rebuttal period, this is still within an acceptable range.

---

> > > ### Author Response · Authors · 2026-04-07
> > >
> > > We would like to thank the reviewer for acknowledging our comprehensive experimental response, and we are dedicated to resolve any remanining concerns the reviewer still has using this additional rebuttal opportunity! In the following, we describe how we will organize LoRA variants into a structured taxonomy in the final revision. Also, given that the number of considered LoRA variants has expanded from 4 to 11 (Tables R5 above and R6 posted under reviewer pJ4H) during the rebuttal, we believe a structured taxonomy is indeed necessary to strengthen our work, and we appreciate the reviewer's valuable suggestion.
> > >
> > > We will categorize the considered variants using a two-layered taxonomy. The first layer separates methods into three categories: **Initialization Variants** and **Architectural Modifications**, which are the distinctions already utilized in the submitted paper, along with a newly introduced category, **Gradient Adjustments**.
> > >
> > > Specifically, *initialization variants* denote methods that vary the initialization of $B$ and $A$, and we further distinguish them by whether the initialization requires training data, yielding data-free and data-informed subcategories. *Architectural modifications* denote methods that alter the forward computation of LoRA, and we further note whether their resulting trainable parameter counts differ from vanilla LoRA's $(m+n)r$ per layer. Finally, *gradient adjustments* denote methods that keep both the structure and initialization identical to vanilla LoRA but modify the gradient computation, e.g., LoRA-Pro that adjusts gradients to approximate full fine-tuning gradient and LoFT that aligns Adam's momentum and variance into the low-rank subspace. With this, we also tried to visualize this taxonomy with respective method assignments:
> > >
> > > **Listing R12** Taxonomy of LoRA variants to be added in the final revision.
> > > ```
> > > LoRA Variants
> > > ├── Initialization Variants
> > > │   ├── Data-free: PiSSA, MiLoRA, Init[AB], OLoRA
> > > │   └── Data-informed: LoRA-GA, EVA
> > > ├── Architectural Modifications
> > > │   ├── Identical param. count: GraLoRA
> > > │   └── Varied param. count: DoRA ($(m+n)r+m$), RandLoRA ($\frac{n^2}{r} + n$)
> > > └── Gradient Adjustments: LoRA-Pro, LoFT
> > > ```
> > > We believe this taxonomy is concise and helps readers understand how the considered methods differ in their optimization mechanisms at a high level. We will still enumerate all methods with their respective algorithms and formulas after the taxonomy has been introduced; if space is insufficient, detailed descriptions will be deferred to the Appendix. We will also highlight several recent LoRA surveys (e.g., [13–14]) as references for readers who wish to dive deeper into the algorithms.
> > >
> > > [13] Mao et al., A survey on lora of large language models. FCS, 2024
> > > [14] Zhu et al., A Survey of LoRA Algorithm Variations for Language Models. NLDB, 2025

---

### Official Review · Reviewer_kugk · 2026-03-08

**Soundness:** 3
**Presentation:** 4
**Significance:** 2
**Originality:** 1
**Overall Recommendation:** 2
**Confidence:** 4

**Summary:**

This paper discusses a major evaluation issue in current LoRA research. It points out that many LoRA variants look better simply because they are compared to a vanilla LoRA baseline that hasn't been properly tuned. By running a thorough grid search on the hyperparameters, the authors show that the actual performance gap is practically negligible, dropping to just 1-2%. The authors use some classic optimization theory to explain why different variants favor different learning rates.

**Compliance With Llm Reviewing Policy:**

Affirmed.

**Final Justification:**

This paper primarily focuses on showing that LoRA performs similarly to its variants. This main claim is valuable and well supported by the numerical experiments.

On the other hand, this paper is very trivial in the sense that all it does is conduct a grid search for LoRA variants. It explains why the optimal learning rates for LoRA variants differ, by citing the well-known theory that step size should be proportional to $1/L$, where $L$ is the smoothness.

The key drawback of this paper is that it does not propose anything novel, just grid search. I think this work is incomplete and can be a very good starting point for a future project. If the authors can provide any theories or methodology about how to tune the step size for LoRA, it will be a very good work. Only suggesting early stopping to prune suboptimal configurations, trying sparser grid search intervals, or using smaller/larger step sizes for some variants, does not contribute much.

**Key Questions For Authors:**

1. Can you include more models at the 7B/8B scale to verify the generalizability of the findings?

2. The current Hessian analysis using the Lanczos algorithm is purely post-hoc and computationally expensive. It does not provide actionable guidance before training. Can you offer a practical heuristic or a closed-form approximation for choosing the optimal step size?

3. DoRA is a primary variant in the empirical evaluation but is explicitly excluded from the Hessian analysis. Could you provide some theoretical intuition or a preliminary numerical analysis of DoRA's loss landscape curvature at initialization?

**Limitations:**

Yes.

**Strengths And Weaknesses:**

Strengths:
1. The paper is well-written, and the presentation of the idea/motivation is clear.
2. The authors conducted extensive experiments across various tasks.

Weakness:
1. The paper primarily focuses on empirical check based on exhaustive hyperparameter grid searches. It does not propose a novel parameter-efficient fine-tuning architecture, nor does it provide a practical, adaptive heuristic for step-size selection.
2. To explain the different optimal learning rates , the authors rely heavily on the classical theory between the optimal learning rate and the maximum Hessian eigenvalue ($\lambda_{max}$) established by LeCun et al. (1992) . This serves merely as an explanation of the empirical results observed. The theoretical contribution of this paper is marginal.
3. The empirical evaluation is restricted to small-scale models, peaking at the 7B parameter scale (Qwen3-0.6B, Gemma-3-1B, and Llama-2-7B). Furthermore, the majority of the experiments are conducted on the small models, with very few results presented at the 7B scale. Without validation on larger foundation models, the core claim that vanilla LoRA universally suffices remains insufficiently supported.

---

> ### Author Rebuttal · Authors · 2026-03-31
>
> We thank the reviewer for the positive feedback on our writing and presentation, as well as the recognition of our experimental scope. We address the reviewer's concerns as follows:
>
> **Weaknesses 1, 2**
>
> We thank the reviewer for the comment regarding originality. While our work does not introduce a new PEFT method or theoretical framework, we believe its originality lies in offering new insights and a more principled understanding of existing methods—an aspect explicitly highlighted in the ICML evaluation criteria. Specifically, per the "ICML 2026 Reviewer Instructions" regarding the judgment of Originality:
>
> > Originality: Does the work provide new insights, deepen understanding, or highlight important properties of existing methods? Does the work introduce new tasks, methods, theory, data, or perspectives that advance the field in some dimensions? Does this work offer a novel combination of existing techniques, and is the reasoning behind this combination well-articulated? Are the contributions clearly distinguished from closely related literature, and is the novelty well justified? **As the questions above indicates, originality does not necessarily require introducing an entirely new method. Rather, a work that provides novel insights by evaluating existing methods, or demonstrates improved understanding is also equally valuable.**
>
> we contend that the submitted paper aligns precisely with this definition of originality through the following:
> (1) a systematic highlight of the acute sensitivity of existing LoRA variants to learning rate tuning
> (2) novel empirical insights demonstrating that vanilla LoRA may suffice for LLM fine-tuning when evaluated under a rigorous and fair hyperparameter tuning protocol
> (3) a demonstration of how classical learning theories (LeCun et al., 1992) offer a robust explanation for the phenomena observed across LoRA and its variants
> (4) a clear distinction of our contributions from closely related literature, as detailed in Section 2.
>
> **Weakness 1 and Question 2**
> We thank the reviewer for raising the point regarding the lack of a practical heuristic for step-size selection. Learning rate tuning is a straightforward practice, with the computational cost scaling only linearly ($N$ times for $N$ candidates), and practitioners can readily monitor validation loss and use early stopping to quickly prune suboptimal ranges. Additionally, users can adopt a sparser search interval to efficiently approximate optimal values, rather than the exhaustive search conducted in our study. We will clarify these practical considerations in the final version. More advanced topics such as efficient hyperparameter tuning [7,8] and the transfer of optimal learning rate across scales [9,10] are important but orthogonal directions beyond the scope of this work.
>
> [7] Bergstra et al., Random search for hyper-parameter optimization, JMLR, 2012
> [8] Falkner et al., BOHB: Robust and Efficient Hyperparameter Optimization at Scale, PMLR, 2018
> [9] Yang et al., Tuning Large Neural Networks via Zero-Shot Hyperparameter Transfer. NeurIPS, 2021
> [10] Chen et al., Learning Rate Scaling across LoRA Ranks and Transfer to Full Finetuning. arXiv, 2026
>
> **Weakness 3 and Question 1**
> We first clarify that the scope of evaluation for Llama-2-7B was consistent with the other two models, already encompassing mathematical reasoning and code generation across rank $r \in$ {8, 32, 128} and batch size $B \in$ {16, 128}. We refer the reviewer to Figure 4, Appendix Figure 8, and Tables 8–13 in the submitted paper for these results.
>
> Per the reviewer's concern on larger models, we have further scaled up our experiments to Llama-2-13B, as presented in Tables R1 and R2 ([posted under reviewer kqpA](https://openreview.net/forum?id=1M64XF14oc&noteId=yyCfpVhJo4)) for rank 8 and rank 128, respectively. The results validate the generalizability of our findings across model scales, where all methods peak at 37.6-40.3% in Table R1 and 41.2-42.7% in Table R2. We also refer the reviewer to Table R5 ([posted under reviewer txFF](https://openreview.net/forum?id=1M64XF14oc&noteId=ROflYav1b9)), where we further validate the generalizability of our findings by including five more LoRA variants and comparing all ten methods on commonsense reasoning tasks.
>
> **Question 3**
> As mentioned in Lines 433–439 (left column) of the paper, we excluded DoRA from Hessian analysis due to its introduction of an additional trainable $m$ (Eq. 2). While DoRA shares the same initialization as LoRA for $B$ and $A$, its unique forward design may affect the training trajectory and Hessian evolution. Yet, when focusing on $B$ and $A$ only, we validate in Table R4 that LoRA and DoRA share identical initial curvature.
>
> **Table R4** $\lambda_{\text{max}}$ of DoRA and LoRA on Query matrix on Qwen-3-0.6B.
> |Trasformer layer |0|1|2|
> |-|-|-|-|
> |**LoRA**|0.381|0.222|0.399|
> |**DoRA**|0.381|0.222|0.399|

---

> > ### Author Rebuttal · Reviewer_kugk · 2026-04-02
> >
> > Thank you for the detailed rebuttal and the additional 13B experiments. This partially resolves my concerns. I admit that your core empirical finding, that LoRA variants perform similarly with proper hyperparameters, is valuable. Your numerical results in the paper and the additional 13B experiments also support this claim well.
> >
> > While I acknowledge the ICML guidelines on originality, the contribution is still not enough.
> >
> > The paper relies entirely on standard grid search and directly applies well-known classical theories to explain the results. The verification step is solid and standard, but it does not advance any theories or methodologies.
> >
> > Your response to the lack of heuristics is not very informative. Using early stopping to prune suboptimal configurations and using sparser search intervals are common practices. So the paper is more like determining a problem with grid search, but not offering any suggestions.
> >
> > Again, I would like to thank the authors for the main claim. It is informative, but the contribution is still limited. If you could propose some new theories or provide some more informative guidence to help with hyperparameter tuning, this would be a very good work.
> >
> > I will maintain my score.

---

> > > ### Author Response · Authors · 2026-04-03
> > >
> > > We thank the reviewer for acknowledging that our core empirical finding is valuable and that our additional 13B numerical results support the claim well. We address the reviewer's remaining concerns as follows:
> > >
> > > **Originality**
> > > We thank the reviewer for acknowledging the guidelines on originality, and we hope the reviewer could consider adjusting the originality evaluation accordingly.
> > >
> > > **Significance**
> > > Per the guideline on Significance:
> > > > Significance: Does the paper address an important or relevant problem? Does it advance understanding, capabilities, or practice in machine learning? Could it influence future research or applications(...)? Is the scope of impact broad or specialized?(...)
> > >
> > > We believe our paper will significantly influence future research: prior to our work, many papers in this area were published in esteemed venues (ICML, ICLR, NeurIPS, NAACL, etc., as shown in Figure 1 and Table 2), but their results might not be well-grounded due to insufficient hyperparameter tuning. **Without our paper, more research will continue to be conducted without proper tuning, and their conclusions will further mislead the community upon acceptance** (as observed in the continuous publication trend in Table 2). We believe our paper can positively influence future research towards greater rigor and comprehensiveness, which is also a significant contribution to this community (as also pointed out by Reviewer txFF).
> > >
> > > We also contend that our study advances the understanding of existing methods. As noted by Reviewer kpqA, while we used a seemingly trivial approach (i.e., hyperparameter tuning) to deliver the core claim, few to none of the earlier studies conducted it carefully, and thus did not uncover the full potential of each method for a fair and reliable comparison. We argue that the simplicity of the approach we apply to verify the core claim cannot be a fair reason for a low significance judgment. In fact, a number of published studies (as summarized in Section 2.1 for various domains) have been conducted to reality-check prior methods through hyperparameter tuning, and they offer deepened understanding of existing methods and provide significant practical insights to the community.
> > >
> > > **Practical heuristics**
> > > We thank the reviewer for the valuable suggestions. During our research, we conclude some practical heuristics that can serve as additional guidance for practitioners, and we will elaborate on them leveraging the additional page:
> > >
> > > + LoRA normally requires larger learning rates compared to most variants, on the order of *e-4. In fact, when batch size is set to 64, LoRA's optimal learning rate falls within this narrow band in 9 out of 10 Tables and Figures provided in the paper and during rebuttal. While the absolute values could vary with models, tasks, ranks, batch sizes, and even training duration, we will suggest readers start exploring learning rates around *e-4 for LoRA.
> > > + Based on the broad experiments, we conclude that a given variant will typically have a fixed relationship (either higher or lower) in optimal learning rates relative to LoRA across different settings. For example, PiSSA requires lower learning rates than LoRA across all experiments, specifically half to one order of magnitude lower in 13 out of 16 Figures and Tables provided in the paper and during rebuttal. Practitioners could leverage the known relationships between methods to better guide the selection of optimal learning rates without explicitly calculating $λ_{max}$. We also highlighted how more LoRA variants differ in optimal learning rate ranges compared to LoRA in the response posted under Reviewer txFF.
> > > + For new methods whose $λ_{max}$ relative to LoRA is not known beforehand, we suggest practitioners estimate the ratio using only a single matrix (e.g., Q_proj at the first layer), as we show in Appendix Figure 11 that the ratios across diverse matrices are consistent (in terms of being either higher or lower to LoRA), rather than performing the complete estimation across all layers as done in Figure 6. This costs only 3–5 minutes on an RTX 3090 and can serve as additional practice to approximate optimal step size efficiently. For instance, Init[AB] has a 2x larger $λ_{max}^{Q, 1}$ compared to LoRA (Figure 11), so one can start exploring learning rates around half of the optimal learning rate of LoRA. Given that LoRA typically works at *e-4 (as stated in bullet point 1 above), practitioners can start from the lower end of *e-4 and possibly down to e-5 (the optimal value is 1.1e-4, as shown in Figure 1).
> > >
> > > We thank the reviewer again for the follow-up comments. We also note that the reviewer's positive feedback on the solidity of our verification step and the presentation of ideas and motivation (i.e., Strength 1) does not appear to be well-reflected in the scores on Soundness (`2: fair`) and Presentation (`2: fair`), and we hope the reviewer could consider adjusting these evaluation scores accordingly.

---

### Official Review · Reviewer_pJ4H · 2026-03-14

**Soundness:** 1
**Presentation:** 2
**Significance:** 3
**Originality:** 2
**Overall Recommendation:** 2
**Confidence:** 5

**Summary:**

The paper investigates whether reported improvements of advanced LoRA variants over vanilla LoRA reflect methodological advances or are artifacts of suboptimal hyperparameter configurations, particularly learning rate tuning. Through a grid searches across Qwen3-0.6B, Gemma-3-1B and Llama-2-7B on math reasoning and code generation, the authors conclude that properly tuned vanilla LoRA matches advanced variants (DoRA, PiSSA, MiLoRA, Init[AB]).

**Compliance With Llm Reviewing Policy:**

Affirmed.

**Final Justification:**

The paper presents a clear and practically valuable empirical finding that careful learning rate tuning largely eliminates performance differences between LoRA variants, but despite the authors’ rebuttal expanding experiments and partially addressing concerns, the evaluation remains insufficiently exhaustive, omitting several recent and fundamentally different methods (LoRA-One, an ICML 2025 oral) to support the breadth of the claims, and therefore I maintain my original assessment.

**Key Questions For Authors:**

see weaknesses

**Limitations:**

yes

**Strengths And Weaknesses:**

# Strengths

1. The paper is clearly structured and easy to follow.

2. The analysis of whether advanced LoRA variants improve over vanilla LoRA is a critical question for the community.

# Weaknesses

1. The current comparison involves only four LoRA variants across two task types and three model sizes. For a paper whose central thesis challenges the validity of a large body of literature, this scope is insufficient to support broad conclusions, ideally it should cover all 52 surveyed methods. Moreover, more recent LoRA variants such as LoRA-Pro, LoFT, LoRA-One, and GaLore represent meaningfully different design (gradient projection, preconditioned updates), and it would be very interesting to see how they perform.

2. The experiments performed at batch size up to 128. This is a low-stochasticity regime. A more informative and challenging demonstration would operate at more stochastic regimes with batch size up to 32 where gradient noise plays a larger role and differences between methods are more likely to be meaningful.

---

> ### Author Rebuttal · Authors · 2026-03-31
>
> We thank the reviewer for the positive feedback on the clarity of our paper and the importance of the research question we address. We respond to the reviewer's concerns as follows:
>
> **Weakness 1**
> We appreciate the reviewer for suggesting a more comprehensive evaluation and for including more variants with fundamentally different designs. Per the reviewer's advice, we scale up our evaluation to model up to the 13B parameter scale (Llama-2-13B), and include one more task type (commonsense reasoning) and five more recently published LoRA variants, resulting in a total coverage of **four models, three task types, and ten LoRA methods (1x vanilla + 9x variants)**.
>
> Specifically, we considered two additional initialization variants (EVA, LoRA-GA), two additional architectural modifications (GraLoRA, RandLoRA), and one gradient adjustment method (LoRA-Pro). The selection for inclusion was solely based on the ease of integration into our existing codebase given the limited rebuttal time (e.g., supported by the PEFT package or well-documented official repositories). These methods represent meaningfully different designs from the methods originally considered in our paper. For example, RandLoRA modifies the gradients of LoRA's $A$ and $B$, GraLoRA splits LoRA into $k$ sub-blocks, and LoRA-GA performs full gradient projection. Moreover, EVA and LoRA-GA require training data to guide initialization, which fundamentally differs from our originally considered data-free initialization variants, i.e., PiSSA, MiLoRA, and Init[AB].
>
> Table R5 ([posted under reviewer txFF](https://openreview.net/forum?id=1M64XF14oc&noteId=ROflYav1b9)) presents the results of all ten methods on Gemma-3-1B fine-tuned on commonsense reasoning tasks, using data released by [6], which consists of 15k training samples over eight QA tasks. The results show that all methods peak at a similar performance level ($\approx$ 37%) and their operating learning rate ranges differ, validating that learning rate matters for diverse LoRA variants. We also refer the reviewer to Tables R1 and R2 ([posted under reviewer kqpA](https://openreview.net/forum?id=1M64XF14oc&noteId=yyCfpVhJo4)), where we present the mathematical reasoning results of Llama-2-13B for rank 8 and rank 128.
>
> While we agree with the reviewer that covering all 52 surveyed methods would be ideal, we argue that doing so would require substantial engineering effort and computation far beyond our available academic resources. The difficulties arise from non-open-sourced or unmaintained codebases, and incompatible packages across implementations—for instance, even the PEFT library supported only around 20 methods. Moreover, what we covered in the submitted paper required over 3,000 independent training runs; with an additional 392 runs conducted during the rebuttal, we are at capacity. Furthermore, some methods introduce their own additional hyperparameters (e.g., $(m, b)$ in BOFT or (init_r, t_init, t_final, delta_T) in AdaLoRA), which would rapidly expand the hyperparameter search space. That said, we plan to open-source our codebase and welcome community efforts to collectively achieve more comprehensive coverage. We also hope the publication of our paper encourages future PEFT research to adopt a more comprehensive hyperparameter tuning protocol on its own.
>
> [6] Hu, et al., Llm-adapters: An adapter family for parameter-efficient fine-tuning of large language models, EMNLP, 2023
>
> **Weakness 2**
> In our submitted paper, we already investigated different stochasticity regimes by tuning batch sizes ($B$) from 16 to 128 in conjunction with learning rates for all methods on various model-task combinations:
> + In Table 1, Gemma-3-1B on math was tuned across $B \in$ {$16, 64, 128$}
> + In Appendix Table 5, Qwen3-0.6B on math was tuned across $B \in$ {$16, 64, 128$}
> + All Llama-2-7B results were tuned across $B \in$ {$16, 128$}, with results presented in  Appendix Figure 8 and Tables 8–13.
>
> A summary of the considered batch size tuning ranges for each model-task combination is available in Appendix Table 3. We also discussed the effect of batch size and its interaction with the optimal learning rate in Section 4.3.1. While our evaluated batch sizes ($B=16, 64, 128$) already cover the stochasticity region suggested by the reviewer ($B=32$), we have expanded Table 1 for LoRA and PiSSA to explicitly include $B=32$ for the reviewer's reference, as shown in Table R3 below. The results confirm that varying batch size does not change our conclusion, as discussed in the submitted paper.
>
> **Table R3** Expanded results of Table 1 with $B=32$ for LoRA and PiSSA under different learning rates.
> |Methods|Batch Size|1.1e-5|2.0e-5|3.6e-5|6.3e-5|1.1e-4|2.0e-4|3.6e-4|6.3e-4|1.1e-3|2.0e-3|3.6e-3|6.3e-3|
> |-|:-:|-|-|-|-|-|-|-|-|-|-|-|-|
> |**LoRA**|**32**|8.61|10.51|11.93|15.22|17.07|19.19|20.57|**20.72**|15.16|9.06|1.33|0.02|
> |**PiSSA**|**32**|12.08|14.78|16.70|19.24|19.43|**20.85**|19.46|16.60|12.76|7.54|2.94|1.67|

---

> > ### Author Rebuttal · Reviewer_pJ4H · 2026-04-03
> >
> > Thank you for the detailed rebuttal and the additional experiments.
> >
> > However, given that the primary contribution is empirical rather than methodological, I believe the strength of the paper should come from the breadth and depth of the evaluation. In its current form, the coverage is still not sufficiently comprehensive to fully support such a general claim across the LoRA literature. In particular, while the inclusion of additional methods and larger models in the rebuttal is a positive step, the space of existing approaches is significantly broader, especially including methods with fundamentally different optimization mechanisms. Similarly, the experimental settings (tasks, stochastic regimes, and hyperparameter dimensions) could be further expanded to ensure the conclusions are robust and not specific to a limited set of conditions.
> >
> > Overall, I view this as a valuable empirical study, but for a venue like ICML, the evaluation and analysis need to be more exhaustive to match the scope of the claim. I will maintain my score.

---

> > > ### Author Response · Authors · 2026-04-07
> > >
> > > We thank the reviewer for the follow-up comments.
> > >
> > > **Clarification of our claim**
> > > We clarify that **we make no general claim across the LoRA literature**. In the submitted paper, we only concluded that limited learning rate exploration may make certain conclusions less robust, and in these contexts, vanilla LoRA may suffice. Of course, we acknowledged that these observations may not generalize fully, that's why we have highlighted several times that our observations may not generalize to untested variants and scenarios (Lines 446–456; Sec. H).
> > >
> > > Our advocates are simple: the follow-up LoRA methods need to carefully tune their own hyperparameters as well as baselines. **Whether or not there exists a method that is generally better than LoRA does not change our advocacy**.
> > >
> > > We would also like to remind the reviewer that our considered tasks and model scales already cover the most common settings in PEFT research (cf. Table 2). Besides, we originally pointed out the scenarios where one might favor a specific variant that marginally outperforms others (Sec. 4.3.2; Lines 427–434, Right).
> > >
> > > **More methods**
> > > During the rebuttal, we have included five additional LoRA variants and illustrated how their mechanisms differ fundamentally. We further expand Table R5 to include OLoRA and LoFT, as shown in Table R6. Results show that out of 11 variants in total, no method provides a clear advantage over vanilla LoRA.
> > >
> > > **Table R6** Expand Table R5 to include two more LoRA variants.
> > > ||2.0e-06|6.3e-06|2.0e-05|6.3e-05|2.0e-04|6.3e-04|2.0e-03|6.3e-03|2.0e-02|
> > > |-|-|-|-|-|-|-|-|-|-|
> > > |**OLoRA**|32.69±.38|36.38±.07|36.64±.14|36.98±.24|**37.17±.19**|36.47±.16|36.91±.08|29.63±10.15|8.97±7.77|
> > > |**LoFT**|7.50±.08|14.25±.35|37.08±.06|37.15±.09|37.38±.13|**37.43±.27**|36.99±.16|31.06±8.93|0.0±.0|
> > >
> > > **More experimental settings**
> > > **a. Tasks**
> > > We further extend to include instruction-following task. We train models on Alpaca and evaluate them by IFEval framework. Table R7 reports the prompt-level strict accuracy.
> > >
> > > **Table R7** Performance on IFEval with Qwen3-0.6B ($r=128,B=64$).
> > > ||2e-06|6.3e-06|2e-05|6.3e-05|2e-04|6.3e-04|2e-03|
> > > |-|-|-|-|-|-|-|-|
> > > |**LoRA**|26.84|26.63|26.93|**28.20**|27.32|26.97|15.30|
> > > |**DoRA**|26.99|26.41|26.41|26.44|**27.42**|26.54|16.86|
> > >
> > > **b. Stochastic regimes**
> > > We further expand $B$ up to 512 in Table R8. With 100k MetaMath training samples, $B=512$ means only 196 optimization steps. The performance of both PiSSA and LoRA starts to decay, with the optimal performance drops from 20–21% (Table 1) to ~17% when $B=512$. Performance degradation associated with large batch sizes is well-documented in the literature [11-12]. Importantly, both methods decay similarly as batch size increases, with neither showing a clear higher peak accuracy within the entire batch size spectrum ($B\in[16,512]$). This confirms again that varying batch size does not change our conclusion, and we will highlight in the Appendix that pushing batch size to extremes can lead to performance degradation.
> > >
> > > **Table R8** Expanded results of Table 1 and Table R3.
> > > ||$B$|1.1e-5|2e-5|3.6e-5|6.3e-5|1.1e-4|2e-4|3.6e-4|6.3e-4|1.1e-3|2e-3|3.6e-3|6.3e-3|
> > > |-|:-:|-|-|-|-|-|-|-|-|-|-|-|-|
> > > |**LoRA**|**256**|4.19|5.62|7.09|9.23|12.05|13.81|16.85|18.12|19.16|**19.29**|0.0|0.0|
> > > ||**512**|2.80|4.05|5.31|6.75|9.19|11.74|14.17|14.24|**16.73**|13.26|0.0|0.0|
> > > |**PiSSA**|**256**|7.73|9.91|12.74|14.31|16.57|18.18|**19.79**|19.71|17.46|15.36|1.90|0.57|
> > > ||**512**|6.25|7.70|10.44|13.26|14.46|16.24|16.78|**17.50**|15.17|8.04|0.77|0.0|
> > >
> > > [11] On Large-Batch Training for Deep Learning: Generalization Gap and Sharp Minima (2017)
> > > [12] Beware of the Batch Size: Hyperparameter Bias in Evaluating LoRA (2026)
> > >
> > > **c. Hyperparameter dimensions**
> > > Beyond learning rate and batch size, we have varied other hyperparameter dimensions (e.g., number of samples in Figure 7). Tables R9 and R10 further vary the number of training epochs and the type of learning rate scheduler, respectively.
> > >
> > > **Table R9** Best performance across different training epochs on math with Gemma-3-1B ($r=128,B=64$).
> > > |Epoch|1|2|3|
> > > |-|-|-|-|
> > > |**LoRA**|20.46±.79|22.71±.33|23.64±.42|
> > > |**DoRA**|20.82±.32|22.69±.70|24.28±.73|
> > > |**Init[AB]**|20.98±.33|23.34±.26|23.98±.38|
> > > |**MiLoRA**|20.03±.59|22.48±.51|23.32±.39|
> > > |**PiSSA**|20.68±.77|22.68±.46|23.37±.20|
> > >
> > > **Table R10** Performance under linear decay scheduler on math with Gemma-3-1B ($r=128,B=64$). Results under cosine annealing is presented in Table 1.
> > > ||6.3e-5|1.1e-4|2.e-4|3.6e-4|6.3e-4|1.1e-3|2.e-3|
> > > |-|-|-|-|-|-|-|-|
> > > |**LoRA**|13.34|15.31|17.16|19.58|**21.07**|20.46|15.98|
> > > |**PiSSA**|17.58|18.94|**20.94**|20.18|20.59|17.04|12.22|
> > >
> > > Collectively, these results suggest that our findings generalize across diverse training setups. Even if some variants were to systematically outperform LoRA under rare hyperparameter settings — which we have not observed in any of our current analyses — it would neither undermine our claim nor invalidate our conclusions and advocacy.

---

### Official Review · Reviewer_kqpA · 2026-03-15

**Soundness:** 3
**Presentation:** 3
**Significance:** 4
**Originality:** 3
**Overall Recommendation:** 5
**Confidence:** 4

**Summary:**

in this paper, the authors perform a reality check on a recent trend of coming up with new variants of LoRA and claiming their supremacy over the older ones. the authors carefully (though, in a sense very simply) test a variety of hyperparameter configurations, with particular focus on the learning rate, and demonstrate that four different variants are in fact .. identical in terms of their resulting accuracies when these optimization hyperparameters were tuned carefully.

in a sense, this is a trivial paper, because the authors of the earlier studies should have followed this when tuning their own hyperparameters. but, clearly none of those earlier studies’ authors did it carefully and misled the community. the authors of the current paper resolve this issue, and i thank the authors.

there is a bit of theoretical-looking analysis connecting the eigenspectrum of the Hessian, but i actually find this section more distracting than not. this section reads like the authors felt like they had to add this section in order to claim that their findings are not trivial, but their main findings are super useful on its own.

thank you, authors!

**Compliance With Llm Reviewing Policy:**

Affirmed.

**Key Questions For Authors:**

no question.

**Limitations:**

see my comments above.

**Strengths And Weaknesses:**

see above.

---

> ### Author Rebuttal · Authors · 2026-03-31
>
> We are glad that the reviewer finds our main findings useful for the community, and we sincerely thank the reviewer for the thoughtful review.
>
> We appreciate the reviewer's remark that the Hessian analysis may feel distracting to readers. We include this analysis to remind the PEFT community that learning rate tuning is a necessary practice, not merely based on the provided empirical observation, but also due to the well-established sharpness-learning rate relationship. Per the reviewer's advice, we will make the transition between the empirical results (Sec. 4) and the Hessian analysis (Sec. 5) smoother in the final revision.
>
> For the reviewer's potential interest, we refer the reviewer to Tables R1, R2 below and Table R5 [posted under reviewer txFF](https://openreview.net/forum?id=1M64XF14oc&noteId=ROflYav1b9), where we further validate the generalizability of our findings across five more LoRA variants (EVA[1], LoRA-GA[2], GraLoRA[3], LoRA-Pro[4], RandLoRA[5]), larger model sizes (Llama-2-13B), and more diverse task types (commonsense reasoning), in response to concerns raised by other reviewers.
>
> **Table R1** Performance of Llama-2-13B on mathematical reasoning tasks with $r=8$ and $B=64$. Both the originally considered five LoRA methods and the newly added LoRA-GA and GraLoRA are presented.
> |Methods|2.00e-05|3.56e-05|6.32e-05|1.12e-04|2.00e-04|3.56e-04| 6.32e-04|1.12e-03|2.00e-03|
> |-|-|-|-|-|-|-|-|-|-|
> |**LoRA**|28.02|30.61|32.80|33.94|36.05|38.29|39.28|**39.70**| 1.45|
> |**DoRA**|28.53|31.29|32.95|33.72|35.55|37.46|39.29|**40.10**| 38.70|
> |**Init[AB]**|26.94|29.10|32.30|34.17|35.44|36.78|39.35|**40.31**|37.28|
> |**MiLoRA**|26.77|30.19|33.68|34.52|35.36|37.20|38.97|**39.22**|0.00|
> |**PiSSA**|30.40|32.41|33.38|34.66|36.12|**38.50**|38.48|37.49| 34.49|
> |**LoRA-GA**|33.16|34.44|37.12|37.36|37.51|**37.62**|36.75|36.07|32.37|
> |**GraLoRA**|26.73|29.06|32.63|34.07|34.75|36.71|38.73|**39.80**|39.52|
>
>
> **Table R2** Performance of Llama-2-13B on mathematical reasoning tasks with $r=128$ and $B=64$.
> |Methods|2.00e-05|3.56e-05|6.32e-05|1.12e-04|2.00e-04|3.56e-04 | 6.32e-04|
> |-|-|-|-|-|-|-|-|
> |**LoRA**|34.70|35.12|37.67|39.47|41.30|**42.23**|0.00|
> |**DoRA** |34.14|35.35|37.21|39.53|41.27|**42.76**| 40.28 |
> |**Init[AB]**|33.70|35.66|36.85|38.95|40.92|**41.62**| 40.27 |
> |**MiLoRA**|32.54|35.23|36.27|38.80|40.84|**41.26**| 39.90 |
> |**PiSSA**|38.18|39.67|**41.17**|41.25|40.83|39.26|34.74|
>
> [1] Paischer et al., Parameter efficient fine-tuning via explained variance adaptation. arXiv, 2024
> [2] Wang et al., LoRA-GA: Low-rank adaptation with gradient approximation. NeurIPS, 2024
> [3] Jung et al., GraLoRA: Granular low-rank adaptation for parameter-efficient fine-tuning. NeurIPS, 2025
> [4] Wang et al., LoRA-Pro: Are low-rank adapters properly optimized? ICLR, 2025
> [5] Albert et al., RandLoRA: Full rank parameter-efficient fine-tuning of large models. ICLR, 2025

---

> > ### Author Rebuttal · Reviewer_kqpA · 2026-04-02
> >
> > .

---

> > > ### Author Response · Authors · 2026-04-07
> > >
> > > We are happy to see all concerns are addressed and we thank the reviewer for the valuable time and insights provided during the rebuttal period.

---

### Decision · Program_Chairs · 2026-04-30

**Decision:**

Reject

**Comment:**

The paper presents a practically valuable empirical study of LoRA variants. The work is technically sound, and the authors provided substantial additional experimental evidence during the rebuttal that addressed the reviewers' concerns. The main claims are well supported. However, this is fundamentally a relatively straightforward empirical assessment of competing approaches, executed competently, but with only modest methodological contributions.